# Rethinking Continual Learning with Progressive Neural Collapse

**Zheng Wang**[1]    **Wanhao Yu**[2]    **Li Yang**[2]    **Sen Lin**[1*]
[1]Department of Computer Science, University of Houston, USA
[2]Department of Computer Science, University of North Carolina at Charlotte, USA

## Abstract

Continual Learning (CL) seeks to build an agent that can continuously learn a sequence of tasks, where a key challenge, namely Catastrophic Forgetting, persists due to the potential knowledge interference among different tasks. On the other hand, deep neural networks (DNNs) are shown to converge to a terminal state termed Neural Collapse during training, where all class prototypes geometrically form a static simplex equiangular tight frame (ETF). These maximally and equally separated class prototypes make the ETF an ideal target for model learning in CL to mitigate knowledge interference. Thus inspired, several studies have emerged very recently to leverage a fixed global ETF in CL, which however suffers from key drawbacks, such as *impracticability* and *limited performance*. To address these challenges and fully unlock the potential of ETF in CL, we propose **Progressive Neural Collapse (ProNC)**, a novel framework that completely removes the need of a fixed global ETF in CL. Specifically, ProNC progressively expands the ETF target in a principled way by adding new class prototypes as vertices for new tasks, ensuring maximal separability across all encountered classes with minimal shifts from the previous ETF. We next develop a new CL framework by plugging ProNC into commonly used CL algorithm designs, where distillation is leveraged to balance between target shifting for old classes and target aligning for new classes. Experiments show that our approach significantly outperforms baselines while maintaining superior flexibility, simplicity, and efficiency. Our code is available at `https://github.com/Continue-Edge-AI-Lab/ProNC`.

## 1 Introduction

Continual Learning (CL) has gained much attention in recent years, aiming to mimic the extraordinary human abilities to learn different tasks in a lifelong manner. A key challenge here is ***Catastrophic Forgetting***, i.e., deep neural networks (DNNs) exhibit a pronounced tendency to lose previously acquired knowledge when trained on new tasks (McCloskey & Cohen, 1989). Within the spectrum of CL scenarios, class-incremental learning (CIL) presents the most formidable setting (Masana et al., 2022), where the model must not only address the current task by differentiating intra-task classes but also retain the knowledge of prior tasks by distinguishing historical classes from newly introduced ones. Yet, achieving this dual objective remains particularly challenging, as evidenced by the suboptimal performance of existing CL methods.

Recent studies (e.g., Papyan et al. (2020)) have identified a compelling empirical phenomenon in DNN training termed ***Neural Collapse (NC)***. During the terminal phase of training—when the training error asymptotically approaches zero—the last-layer features of samples within the same class converge to their class-specific mean, while the means of all classes align with their corresponding classifier prototypes. These prototypes further collapse geometrically to form the vertices of a ***Simplex Equiangular Tight Frame (ETF)***. This phenomenon results in four critical properties: (1) *Feature Collapse*: Features from samples within the same class converge to their class-specific mean, effectively eliminating within-class variability. (2) *ETF Geometric Alignment*: The class-specific means for all classes align with the vertices of an ETF. (3) *Classifier-Prototype Equivalence*: These class means further align with the weights of the linear classifier. (4) *Decision Simplification*: Predictions reduce to a nearest-class-mean rule, where test samples are assigned to the class whose feature mean is the closest.

---

* Corresponding author: slin50@central.uh.edu

A key idea emerging here is that the elegant NC properties of DNN training naturally characterize an ideal model for CL: All classes seen so far will have nearly-zero within-class variability, and their corresponding class-means are equally and perfectly separated. Several studies (Yang et al., 2023b; Dang et al., 2025; Yang et al., 2023a) have recently emerged to leverage NC and predefine a fixed global ETF as the model training target in CL, which however suffer from significant limitations: 1) Setting the number of vertices in the predefined ETF requires the knowledge of total class number encountered during CL before learning the first task, which is clearly not practical. While Yang et al. (2023b) posits that increasing the total class number k can

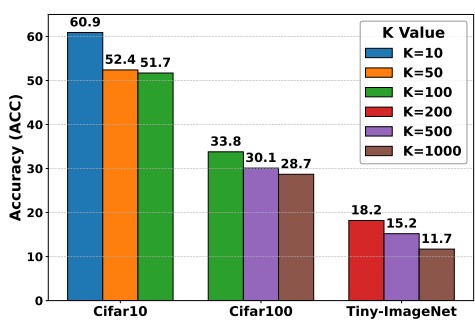

Figure 1: Accuracy under different sizes of predefined ETF in NCT

overcome these limitations, fig. 1 shows that the consequently diminished angle between vertices actually degrades performance. 2) When the total class number is very large, the distance between any two vertices in the predefined ETF will be very small. Pushing class means towards these closely located vertices will unnecessarily hinder class discrimination, especially in early stages of CL when the number of seen classes that the model has to discriminate among is much smaller and the distance between these class means is larger. 3) NC posits that ETFs emerge naturally from feature convergence during training. Predefining the ETF contradicts this emergent behavior, as a randomly initialized ETF risks geometric misalignment between learned features and the imposed topology.

To handle these limitations, a key insight is that the number of vertices in the target ETF for model training should match the total number of classes seen so far to achieve maximal across-class separation anytime in CL. Thus inspired, instead of relying on a predefined ETF with critical design flaws, we seek to develop a novel approach that can appropriately adapt the ETF target for CL in order to fully unleash the potential of NC. To this end, our contributions can be summarized as follows:

*1) A principled approach for ETF expansion.* By rethinking the objective of CL in classification as facilitating progressive NC with a growing ETF after learning each new task, we propose a novel approach, namely ProNC, to dynamically adjust the target ETF during CL. Specifically, ProNC first extracts the initial ETF target that emerges from first task training, and then expands the ETF target by adding new class prototypes as vertices prior to new task learning, to ensure maximal separability across all encountered classes without causing dramatic shifts from the previous ETF. In principle, ProNC can be broadly applied in CL frameworks as a new type of feature regularization.

*2) A simple and flexible framework for CL based on ProNC.* We next develop a new CL framework by plugging ProNC into commonly used CL algorithm designs. In particular, building upon the standard cross-entropy loss for new task learning, we introduce two additional losses, i.e., the alignment loss and the distillation loss. The former seeks to push the learned class features towards the corresponding target ETF provided by ProNC, whereas the later follows a standard idea of knowledge distillation to mitigate feature shifts for old classes. A nearest-ETF classifier will be used to replace the standard linear classifier.

*3) Comprehensive experiments for performance evaluation.* We perform comprehensive experiments on multiple standard benchmarks for both CIL and TIL, to evaluate the effectiveness of our CL approach compared with related baseline approaches. It can be shown that our approach significantly outperforms the baselines especially on larger datasets and also enjoys much less forgetting, without introducing more computation costs. In particular, extensive ablation studies are conducted to justify the benefits of ProNC in terms of maximizing feature separation among different classes and minimizing feature shifts across CL.

## 2 RELATED WORK

**Continual Learning.** In general, existing CL approaches on standard neural networks can be divided into several categories.

*1) Regularization-based approaches* seek to regularize the change on model parameters that are important to previous tasks (Zenke et al., 2017; Chaudhry et al., 2018a). For instance, EWC (Kirkpatrick et al., 2017) penalized updating important weights characterized based on Fisher Information

matrix. MAS (Aljundi et al., 2018) characterized the weight importance based on the sensitivity of model updates if this weight is changed. Liu & Liu (2022) proposed an approach that recursively modified the gradient update to minimize forgetting. The Bayesian framework has also been substantially investigated to implicitly penalize the parameter changes (Lee et al., 2017; Nguyen et al., 2017; Zeno et al., 2021).

*2) Memory-based approaches* store information for previous tasks, which have shown very strong performance and can be further divided into two categories, i,e., rehearsal-based and orthogonal-projection based approaches. Rehearsal-based approaches (Riemer et al., 2018; Chaudhry et al., 2018b; Rolnick et al., 2019) store a subset of data for previous tasks and replay them together with current data for new task learning. Some studies focused on how to select and manage the replay data to achieve better performance and efficiency, such as the use of reservoir sampling (Chrysakis & Moens, 2020), coreset-based memory selection (Borsos et al., 2020; Tiwari et al., 2022; Tong et al., 2025), and data compression (Wang et al., 2022). Some other studies investigated how to utilize the replay data, such as imposing constraints on gradient update (Chaudhry et al., 2018b; Eskandar et al., 2025), combining with knowledge distillation (Rebuffi et al., 2017; Buzzega et al., 2020; Hou et al., 2019; Gao et al., 2022), contrastive learning based approaches (Cha et al., 2021; Wen et al., 2024). The use of generative data has also been explored in CL for rehearsal-based approaches (Shin et al., 2017). Instead of storing data samples, orthogonal-projection based approaches (Farajtabar et al., 2020; Wang et al., 2021; Saha et al., 2021; Lin et al., 2022b;a) store gradient or basis information to reconstruct the input subspaces of old tasks, so as to modify the model parameters only along the direction orthogonal to these subspaces.

*3) Architecture-based approaches* freeze the important parameters for old tasks, train the remaining parameters to learn new tasks and expand the model if needed. Notably, PNN (Rusu et al., 2016) preserved the weights for previous tasks and progressively expanded the network architecture to learn new tasks. LwF (Li & Hoiem, 2017) split the model parameters into two parts, where task-shared parameters are used to extract common knowledge and task-specific parameters are expanded for new tasks. Some studies (Yoon et al., 2017; Hung et al., 2019; Yang et al., 2021) combined the strategies of weight selection, model pruning and expansion.

**Neural Collapse.** The NC phenomenon during the terminal state of DNN training was first discovered in (Papyan et al., 2020), which has further motivated a lot of studies on understanding NC. For example, NC has been investigated under different settings, e.g., imbalanced learning (Yang et al., 2022; Xie et al., 2023), and also been applied in different domains, e.g., semantic segmentation (Shen et al., 2025; Xie et al., 2025) and language models (Wu & Papyan, 2024; Zhu et al., 2024). Very recently, several studies have emerged to leverage NC to facilitate better CL algorithm designs. (Yang et al., 2023a) first proposed to use a fixed global ETF target for feature-classifier alignment in few-shot CL with an ETF classifier, whereas (Yang et al., 2023b) applied the same idea to more general CL setups. (Dang et al., 2025) further integrated this idea with contrastive learning based CL. However, as mentioned earlier, the reliance on a fixed global ETF suffers from critical drawbacks, which we aim to address in this work.

## 3 PRELIMINARIES

**Problem Setup.** We consider a general CL setup where a sequence of tasks $\mathbb{T} = \{t\}_{t=1}^{T}$ arrives sequentially. Each task $t$ is associated with a dataset $\mathbb{D}_t = \{(\mathbf{x}_{t,i}, y_{t,i})\}_{i=1}^{N_t}$ containing $N_t$ input-label pairs. A fixed capacity model parameterized by $\theta$ will be trained to learn one task at a time. This work focuses on two widely studied settings: class-incremental learning (Class-IL) and task-incremental learning (Task-IL). In both settings, there is no overlap in class labels across tasks, ensuring $\mathbb{D}_t \cap \mathbb{D}_{t'} = \emptyset$ for any two distinct tasks $t \neq t'$. In Class-IL, the model does not have task specific information, requiring all data to be classified through a unified global classifier. In Task-IL, task-specific identifier is provided, enabling classification via dedicated task-level classifiers.

**Neural Collapse.** To formally characterize the NC phenomenon (Papyan et al., 2020) emerged during terminal training phases of DNNs, it is necessary to first define the simplex ETF geometry.

**Definition 1** (Simplex Equiangular Tight Frame)**.** *A simplex equiangular tight frame (ETF) is a set of vectors $\{\mathbf{e}_k\}_{k=1}^{K} \in \mathbb{R}^d$ ($d \geq K - 1$) with the following properties. 1) **Equal Norm:** All vectors have identical $\ell_2$-norm, i.e., $\|\mathbf{e}_k\|_2 = 1, \forall k \in \{1, \ldots, K\}$. 2) **Equiangularity:** The inner product between any two distinct vectors is minimal and constant, i.e., $\mathbf{e}_{k_1}^{\top} \mathbf{e}_{k_2} = -\frac{1}{K-1}, \forall k_1 \neq k_2$. Then, a*

*simplex ETF can be constructed from an orthogonal basis $\mathbf{U} \in \mathbb{R}^{d \times K}$ (where $\mathbf{U}^\top \mathbf{U} = \mathbf{I}_K$) via:*

$$\mathbf{E} = \sqrt{\tfrac{K}{K-1}} \mathbf{U} \left( \mathbf{I}_K - \tfrac{1}{K} \mathbf{1}_K \mathbf{1}_K^\top \right), \tag{1}$$

*where $\mathbf{E} = [\mathbf{e}_1, \ldots, \mathbf{e}_K]$ is the ETF matrix, $\mathbf{I}_K$ is the identity matrix, and $\mathbf{1}_K$ is the all-ones vector.*

The NC phenomenon can then be characterized by the following four properties (Papyan et al., 2020): **(NC1)**: The last-layer features of samples within the same class collapse to their within-class mean, resulting in vanishing intra-class variability: the covariance $\boldsymbol{\Sigma}_W^{(k)} \to \mathbf{0}$, where $\boldsymbol{\Sigma}_W^{(k)} = \mathrm{Avg}_i \left\{ (\boldsymbol{\mu}_{k,i} - \boldsymbol{\mu}_k)(\boldsymbol{\mu}_{k,i} - \boldsymbol{\mu}_k)^\top \right\}$, $\boldsymbol{\mu}_{k,i}$ is the feature of sample $i$ in class $k$, and $\boldsymbol{\mu}_k$ is the within-class feature mean of class $k$; **(NC2)**: The centered class means $\{\hat{\boldsymbol{\mu}}_k\}$ align with a simplex ETF, where $\hat{\boldsymbol{\mu}}_k = (\boldsymbol{\mu}_k - \boldsymbol{\mu}_G)/\|\boldsymbol{\mu}_k - \boldsymbol{\mu}_G\|$ and the global mean $\boldsymbol{\mu}_G = \frac{1}{K} \sum_{k=1}^K \boldsymbol{\mu}_k$; **(NC3)**: The centered class means align with their corresponding classifier prototypes, i.e., $\hat{\boldsymbol{\mu}}_k = \mathbf{w}_k/\|\mathbf{w}_k\|$, $1 \le k \le K$, where $\mathbf{w}_k$ is the class prototype of class $k$; **(NC4)**: Under NC1–NC3, predictions reduce to a nearest-class-center rule, i.e., $\arg\max_k \langle \boldsymbol{\mu}, \mathbf{w}_k \rangle = \arg\min_k \|\boldsymbol{\mu} - \boldsymbol{\mu}_k\|$, where $\boldsymbol{\mu}$ is the last-layer feature of a sample for prediction.

# 4 CONTINUAL LEARNING WITH PROGRESSIVE NEURAL COLLAPSE

## 4.1 PROGRESSIVE NEURAL COLLAPSE

To completely remove the need of predefining a global fixed ETF as the feature learning target for CL, we next seek to answer the following two important questions: 1) *How should the base ETF target be initialized?* 2) *How should the ETF target be adapted during CL?*

*1) ETF initialization after first task.* Previous studies (Yang et al., 2023b) randomly initialize the ETF target, which could lead to potential misalignment between the predefined ETF and learned features during task learning. Note that after the training of Task 1, last-layer class feature means $\{\boldsymbol{\mu}_c\}_{c=1}^{K_1}$ converge to an ETF $\mathbf{E}^{d \times K_1}$, where $\boldsymbol{\mu}_c \in \mathbb{R}^d$ and $K_1$ is the number of classes in Task 1. Thus motivated, the initial ETF should be extracted from the first task training to address the misalignment, which leads to an ETF target that matches the number of classes in Task 1.

However, in practice it is difficult to fully reach the asymptotic convergence regime of model training with zero training loss, such that the learned class feature means $\tilde{M}_{K_1} = \{\tilde{\boldsymbol{\mu}}_c\}_{c=1}^{K_1}$ for Task 1 will not strictly satisfy the ETF properties, as also corroborated in our empirical observations. Here $\tilde{\boldsymbol{\mu}}_c$ is the empirical feature mean over samples within the class $c$. To handle this, a key step is to find the right ETF target that is closest to $\tilde{M}_{K_1}$ after the training for Task 1 converges, i.e., $\mathbf{E}^* = \arg\min_{\mathbf{E}} \|\tilde{M}_{K_1} - \mathbf{E}\|_F^2$. To this end, based on Definition 1, we have the following theorem that characterizes the nearest ETF after Task 1:

**Theorem 1.** *Let $\mathbf{U}' = \sqrt{\frac{K_1-1}{K_1}} \tilde{M}_{K_1} \left( \mathbf{I}_{K_1} - \frac{1}{K_1} \mathbf{1}_{K_1} \mathbf{1}_{K_1}^\top \right)$ and the SVD of $\mathbf{U}'$ is $\mathbf{W}\boldsymbol{\Sigma}\mathbf{V}^\top$. Then the ETF matrix $\mathbf{E}^*$ can be obtained as follows:*

$$\mathbf{E}^* = \sqrt{\tfrac{K_1}{K_1-1}} \mathbf{W}\mathbf{V}^\top \left( \mathbf{I}_{K_1} - \tfrac{1}{K_1} \mathbf{1}_{K_1} \mathbf{1}_{K_1}^\top \right). \tag{2}$$

Given the learned class feature means $\tilde{M}_{K_1}$, Theorem 1 immediately indicates a three-step procedure to construct the initial ETF target that aligns well with the task learning: 1) Construct $\mathbf{U}'$ from $\tilde{M}_{K_1}$, 2) Conduct SVD on $\mathbf{U}'$, 3) Construct $\mathbf{E}^*$ based on Equation (2).

*2) ETF expansion prior to new task learning.* Given the initial ETF matrix $\mathbf{E}_1 = \mathbf{E}^*$ where the number of vertices matches the number of classes in Task 1, the next step is to progressively expand the ETF target as new classes come in with new tasks, in order to achieve two objectives: 1) the number of newly expanded vertices in the new ETF target will match the number of new classes in the new task; 2) the vertices in the new ETF target that match old classes will not significantly shift from their original positions in the old ETF target, so as to reduce catastrophic forgetting.

A key insight here is that the ETF matrix $\mathbf{E}$ is indeed determined by the corresponding orthogonal basis $\mathbf{U}$ as shown in Equation (1), and keeping the orthogonal basis unchanged when expanding the ETF will in principle reduce the shift from the old ETF. This motivates a novel ETF expansion strategy by expanding the constructing orthogonal basis, which includes two steps:

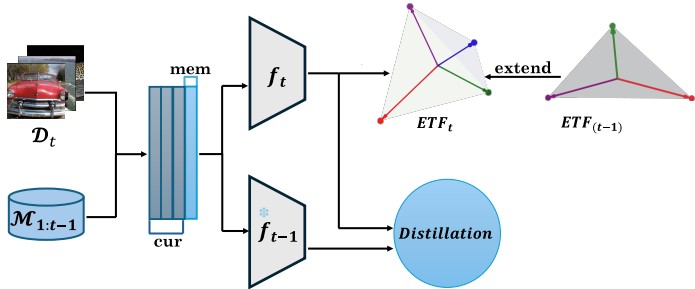

Figure 2: Overflow of our proposed CL framework for new task learning based on the mixture of current task data and replay data. The new model $f_t$ is trained towards the expanded ETF target, with forgetting further reduced based on feature distillation.

*(Step a)* Let $K_t$ be the total number of classes until any task $t$. When a new task $t \geq 2$ arrives with $K_t - K_{t-1}$ classes, the ETF target expansion will be triggered before learning task $t$, which seeks to obtain a new target $\mathbf{E}_t$ with $K_t$ vertices from the previous ETF target $\mathbf{E}_{t-1}$ with $K_{t-1}$ vertices. In particular, the original orthogonal basis $\mathbf{U}_{t-1} \in \mathbb{R}^{d \times K_{t-1}}$ of $\mathbf{E}_{t-1}$ will be expanded to $\mathbf{U}_t \in \mathbb{R}^{d \times K_t}$ by appending $K_t - K_{t-1}$ new orthogonal vectors. These new vectors are generated via Gram-Schmidt orthogonalization against the existing $K_{t-1}$ basis vectors in $\mathbf{U}_{t-1}$, ensuring that $\mathbf{U}_t$ retains orthonormality across all $K_t$ vectors.

*(Step b)* Substituting $\mathbf{U}_t$ and $K_t$ into Equation (1) will lead to an expanded ETF target with $K_t$ vertices. This extended ETF serves as the predefined geometric configuration for feature learning in task $t$, maintaining uniform angular separation and maximal equiangularity among all seen classes.

## 4.2 A CONTINUAL LEARNING FRAMEWORK BASED ON PROGRESSIVE NEURAL COLLAPSE

In what follows, we seek to incorporate the idea of progressive neural collapse (ProNC) into commonly used CL algorithm designs, where the model will be trained to push the learned class features towards the progressively expanded ETF for each task during CL. More specifically, we focus on the model training for tasks $t \geq 2$, whereas the first task learning follows a standard supervised learning procedure with widely used loss functions, e.g., cross-entropy loss $\mathcal{L}_{\text{ce}}$. For tasks $t \geq 2$, we first apply ProNC to generate a newly expanded ETF target $\mathbf{E}_t$ before learning task $t$. The loss function design for model training will include three different loss terms, i.e., a **supervised term**, an **alignment term**, and a **distillation term**. The first supervised term follows the standard cross-entropy loss to facilitate intra-task classification, while we introduce the other two loss terms below in detail.

*1) Alignment with the ETF target.* The alignment loss pushes the learned class features towards the ETF target $\mathbf{E}_t = [\mathbf{e}_{1,t}, ..., \mathbf{e}_{K_t,t}]$, which characterizes the cosine similarity between the feature $\boldsymbol{\mu}_{k,i}$ for sample $i$ in class $k$ and the corresponding vertex $\mathbf{e}_{k,t}$ in the ETF $\mathbf{E}_t$ for class $k$ (Yang et al., 2022; 2023b):

$$\mathcal{L}_{\text{align}}(\boldsymbol{\mu}_{k,i}^t, \mathbf{e}_{k,t}) = \tfrac{1}{2}(\mathbf{e}_{k,t}^\top \boldsymbol{\mu}_{k,i}^t - 1)^2. \tag{3}$$

Here $\boldsymbol{\mu}_{k,i}^t$ corresponds to the normalized feature extracted from the last layer of the current model when learning task $t$. A small $\mathcal{L}_{\text{align}}$ implies that the learned normalized feature for each sample aligns well the corresponding ETF vertex for the class that the sample belongs to, minimizing the intra-class variability while forcing different class feature means equally separated. This indeed provides a new type of feature regularization based on our progressively expanded ETF, which can be widely adopted and is also very powerful as shown later in our experimental results.

*2) Distillation to further reduce forgetting.* While ProNC expands the ETF target without dramatically shifting from the old ETF, the vertices that map to old classes will inevitably change after ETF expansion due to geometric properties of ETF. To handle this and further reduce catastrophic forgetting, we next borrow the idea of knowledge distillation, which is a widely used technique in CL to reduce catastrophic forgetting (Rebuffi et al., 2017; Hou et al., 2019; Buzzega et al., 2020; Yang et al., 2023b). In particular, we consider a typical distillation loss which characterizes the cosine similarity between the normalized features, for the same data sample, from the current model and that from the model obtained after learning the previous task, to maintain the simplicity and flexibility of our framework:

$$\mathcal{L}_{\text{distill}}(\boldsymbol{\mu}_{k,i}^{(t-1)}, \boldsymbol{\mu}_{k,i}^{(t)}) = \tfrac{1}{2}((\boldsymbol{\mu}_{k,i}^{(t-1)})^\top \boldsymbol{\mu}_{k,i}^{(t)} - 1)^2. \tag{4}$$

Here $\boldsymbol{\mu}_{k,i}^{(t-1)}$ is the normalized last layer feature for the sample $i$ in class $k$ after task $t-1$.

To best unlock the potential of ProNC, we also leverage the standard data replay in the CL framework: the replay data from previous tasks will be mixed together with the current data before the current task learning, such that each minibatch of data during model training will include both replay data and current data. Combining all three loss terms will lead to the final instance-loss function:

$$\mathcal{L} = \mathcal{L}_{\text{ce}} + \lambda_1 \cdot \mathcal{L}_{\text{align}} + \lambda_2 \cdot \mathcal{L}_{\text{distill}}, \tag{5}$$

which is averaged over all samples in a minibatch for model update. Here $\lambda_1$ and $\lambda_2$ are coefficients to balance between pushing the learned features towards the target ETF and reducing the vertex shift from the previous ETF for minimizing forgetting. The overall workflow is shown in Figure 2.

**Inference.** During inference for model performance evaluation, instead of using the linear classifier for classification, we leverage the widely used cosine similarity (Peng et al., 2022; Gidaris & Komodakis, 2018; Wang et al., 2018; Hou et al., 2019) between the extracted sample feature from the learned model and the vertices in the target ETF as the classification criteria. Specifically, for any testing data sample $i$, let $\boldsymbol{\mu}_j$ denote the normalized last layer feature extracted from the tested model. Then the class prediction result will be $\arg\max_k \boldsymbol{\mu}_j^\top \boldsymbol{e}_k$, where $\boldsymbol{e}_k$ is the $k$-th column vector in the target ETF.

# 5 EXPERIMENTAL RESULTS

## 5.1 EXPERIMENTAL SETUPS

***Datasets and baseline approaches.*** We evaluate the performance on three standard CL benchmarks, i.e., **Seq-CIFAR-10** (Krizhevsky et al., 2009) that partitions 10 classes into 5 sequential tasks, **Seq-CIFAR-100** (Krizhevsky et al., 2009) that comprises 10 tasks with 10 classes per task, and **Seq-TinyImageNet** (Le & Yang, 2015) divides 200 classes into 10 tasks, and consider both Class-IL and Task-IL scenarios.

We evaluate our CL approach against state-of-the-art (SOTA) approaches, including various replay-based approaches, i.e., ER (Riemer et al., 2019), iCaRL (Rebuffi et al., 2017), GEM (Lopez-Paz & Ranzato, 2017), GSS (Aljundi et al., 2019), DER (Buzzega et al., 2020), DER++ (Buzzega et al., 2020), STAR (Eskandar et al., 2025), CSReL (Tong et al., 2025), and contrastive learning based approaches, i.e., $\text{Co}^2\text{L}$ (Cha et al., 2021), CILA (Wen et al., 2024), $\text{MNC}^3\text{L}$ (Dang et al., 2025). We also compare our approach with NCT (Yang et al., 2023b) that predefines a fixed global ETF target.

***Implementation details and evaluation metrics.*** To ensure a fair comparison, we train a ResNet-18 backbone (He et al., 2016) using the same number of epochs and batch size for all approaches. For contrastive learning approaches $\text{Co}^2\text{L}$, CILA, and $\text{MNC}^3\text{L}$, we follow their original implementations by removing the final classification layer of ResNet-18, and append a two-layer projection head with ReLU activation to map backbone features into a $d$-dimensional embedding space ($d = 128$ for Seq-CIFAR-10/100, $d = 256$ for Seq-TinyImageNet). The training epochs are 50 for Seq-CIFAR-10 and Seq-CIFAR-100, and 100 for Seq-TinyImageNet instead of 500 epochs for the initial tasks, 100 (Seq-CIFAR-10 and Seq-CIFAR-100) and 50 (Seq-TinyImageNet) for incremental tasks. A separate linear classifier is subsequently trained on the frozen embeddings for these contrastive learning methods. For STAR ( Eskandar et al. (2025)), we employ the performance of ER+STAR, since our method can be seen as adding a regularization term to ER. Hyperparameter details are in Appendix A.2. We consider two standard evaluation metrics in CL, i.e., final average accuracy (FAA) and average forgetting (FF). Let $T$ denote the total number of tasks and $a_i^t$ be the model accuracy on the $i$-th task after learning the task $t \in [1, T]$. The FAA and FF are defined as:

$$\text{FAA} = \frac{1}{T} \sum_{i=0}^{T-1} a_i^{T-1}, \ \text{FF} = \frac{1}{T-1} \sum_{i=0}^{T-2} \max_{t \in \{0, \ldots, T-2\}} a_i^t - a_i^{T-1}.$$

## 5.2 MAIN RESULTS

Table 1 shows the performance comparison for both Class-IL and Task-IL under 200 and 500 memory budgets. It is clear that our approach significantly and consistently outperforms all the baseline approaches across all considered CL settings, datasets, and buffer sizes. In particular, the performance improvement in our approach becomes more substantial on larger datasets and under a smaller buffer size. For example, consider a buffer size of 200. On Seq-CIFAR-100, our approach

Table 1: Performance comparison under various setups. All results are averaged over multiple runs. The final version with error bars is in the appendix.

| Buffer | Method | Seq-CIFAR-10 | | Seq-CIFAR-100 | | Seq-TinyImageNet | |
|---|---|---|---|---|---|---|---|
| | | Class-IL | Task-IL | Class-IL | Task-IL | Class-IL | Task-IL |
| | | FAA (FF) | FAA (FF) | FAA (FF) | FAA (FF) | FAA (FF) | FAA (FF) |
| 200 | ER (Riemer et al., 2019) | 44.79 (59.30) | 91.19 (6.07) | 21.78 (75.06) | 60.19 (27.38) | 8.49 (76.53) | 38.17 (40.47) |
| | iCaRL (Rebuffi et al., 2017) | 49.02 (23.52) | 88.99 (25.34) | 28 (47.20) | 51.43 (36.20) | 7.53 (31.06) | 28.19 (42.47) |
| | GEM (Lopez-Paz & Ranzato, 2017) | 25.54 (80.36) | 90.44 (9.57) | 20.75 (77.40) | 58.84 (29.59) | – | – |
| | GSS (Aljundi et al., 2019) | 39.07 (72.48) | 88.8 (8.49) | 19.42 (77.62) | 55.38 (32.81) | – | – |
| | DER (Buzzega et al., 2020) | 61.93 (35.79) | 91.4 (6.08) | 31.23 (62.72) | 63.09 (25.98) | 11.87 (64.83) | 40.22 (40.43) |
| | DER++ (Buzzega et al., 2020) | 64.88 (32.59) | 91.92 (5.16) | 28.13 (60.99) | 66.80 (23.91) | 11.34 (73.47) | 43.06 (39.02) |
| | LODE (Liang & Li, 2023) | 68.01 (24.63) | 93.11 (4.75) | 26.65(44.29) | 71.23 (18.75) | 15.13 (64) | 51.42 (29.66) |
| | Co$^2$L (Cha et al., 2021) | 51.27 (30.17) | 84.69 (2.91) | 18.09 (64.14) | 49.19 (27.83) | 12.95 (62.04) | 38.40 (40.75) |
| | CILA (Wen et al., 2024) | 59.68 (37.52) | 91.36 (5.89) | 19.49 (64.01) | 53.93 (33.07) | 12.98 (63.11) | 37.32 (41.40) |
| | MNC$^3$L (Dang et al., 2025) | 51.09 (33.74) | 85.07 (4.90) | 15.81 (62.51) | 43.91 (39.79) | 10.57 (59.68) | 32.78 (45.10) |
| | STAR (Eskandar et al., 2025) | 65.94 (15.99) | 95.12 (2.06) | 38.15 (42.17) | 79.53 (17.32) | 13.64 (68.51) | 43.01 (39.16) |
| | CSReL (Tong et al., 2025) | 37.46 (26.34) | 69.22 (17.16) | 29.06 (58.23) | 66.99 (23.20) | 18.14 (49.77) | 45.04 (34.12) |
| | NCT (Yang et al., 2023b) | 51.59 (22.48) | 80.63 (1.41) | 26.38 (27.40) | 75.75 (4.79) | 10.95 (49.33) | 52.71 (15.88) |
| | **Ours** | **65.58 (32.75)** | **96.86 (0.65)** | **42.99 (36.07)** | **85.63 (4.40)** | **27.44 (42.81)** | **69.09 (9.42)** |
| 500 | ER (Riemer et al., 2019) | 57.74 (43.22) | 93.61 (3.50) | 22.35 (73.08) | 73.98 (16.23) | 9.99 (75.21) | 48.64 (30.73) |
| | iCaRL (Rebuffi et al., 2017) | 47.55 (28.20) | 88.22 (22.61) | 33.25 (40.99) | 58.16 (27.90) | 9.38 (37.30) | 31.55 (39.44) |
| | GEM (Lopez-Paz & Ranzato, 2017) | 26.2 (78.93) | 92.16 (5.60) | 25.54 (71.34) | 66.31 (20.44) | – | – |
| | GSS (Aljundi et al., 2019) | 49.73 (59.18) | 91.02 (6.37) | 21.92 (74.12) | 60.28 (26.57) | – | – |
| | DER (Buzzega et al., 2020) | 70.51 (24.02) | 93.40 (3.72) | 41.36 (49.07) | 71.73 (25.98) | 17.75 (59.95) | 51.78 (28.21) |
| | DER++ (Buzzega et al., 2020) | 72.70 (22.38) | 93.88 (4.66) | 38.20 (49.18) | 74.77 (15.75) | 19.38 (58.75) | 51.91 (25.47) |
| | LODE (Liang & Li, 2023) | 75.91 (18.18) | 94.19 (3.94) | 40.01 (32.58) | 80.06 (8.96) | 20.5 (56.51) | 61.49 (18.61) |
| | Co$^2$L (Cha et al., 2021) | 61.78 (17.79) | 89.51 (2.65) | 26.64 (48.60) | 62.32 (23.47) | 18.71 (49.64) | 50.74 (13.80) |
| | CILA (Wen et al., 2024) | 67.82 (18.22) | 93.29 (0.65) | 31.27 (45.67) | 68.29 (16.98) | 18.09 (64.14) | 49.19 (27.83) |
| | MNC$^3$L (Dang et al., 2025) | 52.20 (27.88) | 85.94 (3.15) | 22.29 (46.09) | 56.43 (24.29) | 11.52 (44.96) | 36.32 (39.00) |
| | STAR (Eskandar et al., 2025) | 69.19 (12.21) | 95.36 (2.38) | 47.56 (28.68) | 78.28 (15.43) | 21.31 (57.2) | 59.32 (19.49) |
| | CSReL (Tong et al., 2025) | 40.82 (29.18) | 73.30 (16.89) | 39.22 (47.03) | 75.09 (15.55) | 22.13 (50.01) | 51.94 (28.20) |
| | NCT (Yang et al., 2023b) | 60.93 (13.82) | 81.27 (3.84) | 33.77 (27.85) | 75.87 (5.06) | 18.24 (50.90) | 62.30 (6.84) |
| | **Ours** | **73.95 (26.75)** | **96.95 (0.24)** | **48.94 (24.32)** | **86.38 (4.35)** | **29.06 (38.58)** | **69.77 (9.52)** |

outperforms the best baseline approaches, i.e., DER for Class-IL and NCT for Task-IL, by 37.65% and 13.04%, respectively. On Seq-TinyImageNet, our approach outperforms the best baseline approaches, i.e., CSREL for Class-IL and NCT for Task-IL, by 59.32% and 31.08%, respectively. In particular, the performance of our approach is outstanding when the buffer size is 200, especially for Task-IL when we focus on differentiating classes within a specific task. This implies that our approach is more robust to different buffer sizes and the principle of ProNC can even work well with a small amount of replay data. Moreover, by leveraging the NCT in a more principled manner, our approach dominates the previous approach NCT where a predefined global ETF target hinders class discrimination by unnecessarily forcing class means towards closely located vertices.

Besides, it can be seen from Table 1 that among most scenarios, our approach and NCT achieve much less forgetting compared to other baseline approaches, while our approach shows even better forgetting than NCT in 8 out of 12 settings. The reason is that, instead of simply constraining the shifts from old features or important weights as in previous studies, the ETF target offers an additional fixed goal from which the model should not shift the new features too far away. These results indicate the huge potential of leveraging NC and ETF in guiding the feature learning for CL.

More interestingly, in contrast to previous replay-based approaches (Riemer et al., 2019; Rebuffi et al., 2017; Lopez-Paz & Ranzato, 2017; Aljundi et al., 2019; Buzzega et al., 2020) which re-

Table 2: Performance comparison with buffer size zero.

| Buffer | Method | Seq-CIFAR-100 | | Seq-TinyImageNet | |
|---|---|---|---|---|---|
| | | Class-IL | Task-IL | Class-IL | Task-IL |
| 0 | Co$^2$L (Cha et al., 2021) | 26.06 (68.82) | 51.91 (40.02) | 13.43 (65.75) | 38.98 (40.77) |
| | MNC$^3$L (Dang et al., 2025) | 30.48 (53.03) | 56.69(37.02) | 14.04 (54.25) | 42.59 (37.89) |
| | **Ours** | **32.28 (45.92)** | **84.62 (4.39)** | **24.43 (46.14)** | **68.08 (9.81)** |

quire a replay buffer in principle, *our approach can even work without replay.* In table 2, we report the results for both Class-IL and Task-IL under an empty memory buffer. Compared with contrastive-learning-based methods such as Co2L (Cha et al., 2021) and MNC3L (Dang et al., 2025), our method still achieves superior performance. Specifically, with an *empty* replay buffer, our approach can achieve a FAA of 32.28% for Class-IL and 84.62% for Task-IL on Seq-CIFAR-100, and 24.43% for Class-IL and 68.08% for Task-IL on Seq-TinyImageNet. Remarkably, these results almost surpass all baselines in table 5 even when those baselines use a buffer size of 200 on Seq-CIFAR-100, and for Seq-TinyImageNet, without a memory buffer, ProNC outperforms all baselines using buffer sizes of both 200 and 500. *This phenomenal performance implies that our approach indeed offers a new and powerful feature regularization based on ProNC, which can be widely applied in various CL scenarios.*

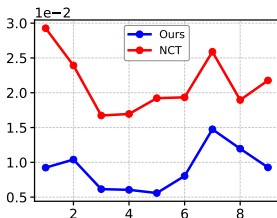

(a) Cosine similarity of feature means between any two different classes, averaged over all seen classes.

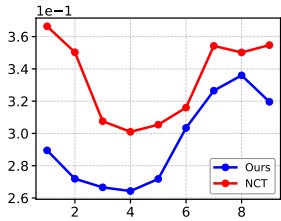

(b) Std of cosine similarity of feature means between any two different classes in the same task, averaged over tasks.

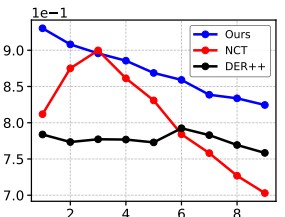

(c) Cosine similarity between best previous features and final features for the same class, averaged over all seen classes.

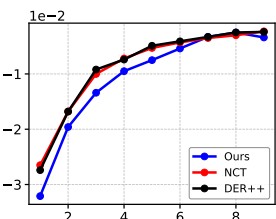

(d) Cosine similarity between one class feature mean and the classifier prototype of any seen different class.

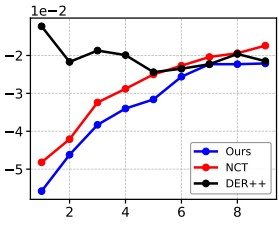

(e) Cosine similarity between one class feature mean and the classifier prototype of another class in the same task.

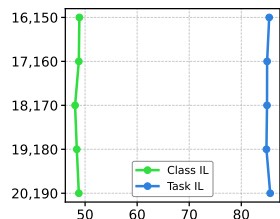

(f) Performance robustness of our approach under different values of the hyperparameters $(\lambda_1, \lambda_2)$.

Figure 3: In (a)-(e), X-axis is the task ID during CL and Y-axis is the (average or std) value of the corresponding cosine similarity. In (f), X-axis is the accuracy FAA and Y-axis is the value of $(\lambda_1, \lambda_2)$.

## 5.3 ABLATION STUDY

In what follows, we will conduct various ablation studies to build a comprehensive understanding of our approach, where most studies are for Class-IL on Seq-CIFAR-100 with a buffer size 500.

*Feature learning behaviors.* To understand the superior performance of our approach in contrast to important baseline approaches, we first delve into the feature learning behaviors during CL by characterizing different types of feature correlations. In particular, based on our analysis of the ETF target and Table 1, compared to NCT (Yang et al., 2023b) with a predefined global ETF, our approach should enjoy the following benefits:

1) A lower cosine similarity between different class feature means, which is also closer to the theoretically maximum separation $-\frac{1}{K_t-1}$ for $K_t$ seen classes until task $t$ according to Definition 1. This can be confirmed in Figure 3a, where we calculate $\mathrm{Avg}_{k\neq k'}(\langle \frac{\mu_k-\mu_G}{||\mu_k-\mu_G||}, \frac{\mu_{k'}-\mu_G}{||\mu_{k'}-\mu_G||}\rangle + \frac{1}{K_t-1})$ of the features after learning every task $t$, averaged for any two seen classes $k$ and $k'$. Thanks to ProNC, our approach achieves a smaller value, which is also closer to 0, than NCT. In NCT, with a global ETF of $K_{\mathrm{global}}$ classes, the pairwise inner product between any two normalized ETF vertices is $-1/(K_{\mathrm{global}} - 1)$, which is very close to 0 when $K_{\mathrm{global}}$ is large (e.g., $-1/999$ for $K_{\mathrm{global}} = 1000$), so early-task classes are packed with relatively small angular separation. 2) All class feature means should be

Table 3: ProNC as feature regularization

| Method | FAA |
|---|---|
| iCaRL (Rebuffi et al., 2017) | 33.25 |
| **iCaRL with ProNC** | **38.87** |
| LUCIR (Hou et al., 2019) | 37.68 |
| **LUCIR with ProNC** | **40.80** |
| ER (Riemer et al., 2019) | 22.35 |
| ER with LODE (Liang & Li, 2023) | 35.21 |
| ER with STAR (Eskandar et al., 2025) | 26.77 |
| **ER with ProNC** | **48.94** |
| DER++ (Buzzega et al., 2020) | 38.20 |
| DER++ with LODE (Liang & Li, 2023) | 40.01 |
| DER++ with STAR (Eskandar et al., 2025) | 39.77 |
| **DER++ with ProNC** | **47.34** |
| XDER (Buzzega et al., 2020) | 49.93 |
| **XDER with ProNC** | **51.32** |

almost equally separated within the same task. To show this, we evaluate the standard deviation (std) of across-class cosine similarity for the same task, averaged over all seen tasks in CL, i.e., $\mathrm{Avg}(\mathrm{std}(\langle \frac{\mu_k-\mu_G}{||\mu_k-\mu_G||}, \frac{\mu_{k'}-\mu_G}{||\mu_{k'}-\mu_G||}\rangle))$, where classes $k$ and $k'$ are in the same task. Figure 3b shows that our approach achieves a smaller average std (also closer to 0) than NCT, implying the different class means are closer to equal separation. Besides, we also follow the metrics in NCT (Yang et al.,

2023b) to evaluate 1) the cosine similarity between the feature mean of class $k$ and the classifier prototype $w_{k'}$ of a different class $k'$, averaged over all seen classes until task $t$. 2) the same cosine similarity but for classes within the same task, which is averaged over all seen tasks during CL. As shown in Figure 3d and 3e, our approach achieves a lower cosine similarity compared to NCT and DER++, which implies easier class discrimination.

Moreover, Table 1 demonstrates the exceptional performance of our approach in addressing forgetting. To understand this, we characterize the cosine similarity between 1) the learned features from the best performing previous model and 2) that from the final model, for the same class, averaged over all seen classes so far. This is consistent to the definition of forgetting. As shown in Figure 3c, our approach achieves a higher similarity than both NCT and DER++, indicating its superior capability in handling feature shifts during CL and then minimizing forgetting based on ProNC.

*Generality of our approach.* As discussed in Section 5.2, our approach introduces a novel and effective feature regularization method based on ProNC, which can be generally incorporated into different CL frameworks. To further verify this, we conduct more experiments by plugging ProNC into established replay-based methods (iCaRL (Rebuffi et al., 2017), LUCIR (Hou et al., 2019)), ER (Riemer et al., 2019), DER++ (Buzzega et al., 2020), and XDER (Buzzega et al., 2020). For studies with feature-wise distillation (iCaRL and LUCIR), we keep all

Table 4: Impact of different loss design

| Loss Combination | FAA |
|---|---|
| $\mathcal{L}_{\text{align}} + \mathcal{L}_{\text{distill}}$ | 48.94 |
| $\mathcal{L}_{\text{align}} + l_2$-Norm loss | 49.42 |
| $l_2$-Norm loss $+ \mathcal{L}_{\text{distill}}$ | 48.61 |
| $l_2$-Norm loss $+ l_2$-Norm loss | 48.66 |

the components the same as the original designs in these studies (including the classifier design) and only incorporate ProNC with the alignment loss. For studies without feature-wise distillation (ER and DER++), in addition to the alignment loss, we also add feature-wise distillation and keep all the original components. Table 3 shows the performance comparison among iCaRL, LUCIR, ER, DER++, XDER, and the versions enhanced with the ProNC regularization. For the SOTA regularization STAR (Eskandar et al., 2025) and LODE Liang & Li (2023), the training of current and buffer data needs to be separated, making it unsuitable for iCaRL and LUCIR. Clearly, incorporating the ProNC regularization yields substantial performance improvements over both the original design, LODE, and STAR, which proves the potential and generality of ProNC as a feature regularization for CL.

*Flexibility of our designs.* Our approach is very flexible in the sense that the loss terms in the final objective function can be replaced by other designs. To show this, we replace the cosine-similarity in the loss functions, i.e., $\mathcal{L}_{\text{align}}$ and $\mathcal{L}_{\text{distill}}$, by using a standard $l_2$-norm, and conduct experiments under the Class-IL setting on Seq-CIFAR-100 with a memory buffer size of 500. As shown in Table 4, the performance of our approach is stable under different design combinations when we replace the cosine similarity in any of the two loss terms, which further corroborate the

Table 5: Impact of different components.

| Variant | Performance |
|---|---|
| Ours | 48.94 |
| (a) w/o $\mathcal{L}_{\text{ce}}$ | 44.97 |
| (b) w/o $\mathcal{L}_{\text{align}}$ | 23.22 |
| (c) w/o $\mathcal{L}_{\text{distill}}$ | 19.96 |
| (d) w/ predefined base ETF | 44.99 |
| (e) w/ predefined global ETF | 33.51 |
| (f) w/ linear classifier | 44.49 |

flexibility of our approach. In principle, our approach can be generally applied with a loss function that seeks to minimize the distance between the learned features and the ETF target/old features.

*Impacts of different components.* To understand the impact of different design components on our approach, we investigate six different variants, as shown in Table 5. Here in (a)-(c) we remove one of the three loss terms in Equation (5), respectively. Clearly, removing either $\mathcal{L}_{\text{align}}$ or $\mathcal{L}_{\text{distill}}$ will significantly degrade the performance, highlighting the importance of the designed ETF target and the right balance between learning stability and plasticity. On the other hand, while removing $\mathcal{L}_{\text{ce}}$ will slightly decrease the performance, this variant still outperforms the baseline approaches as shown in Table 1, indicating the benefit of our approach mainly from $\mathcal{L}_{\text{align}}$ and $\mathcal{L}_{\text{distill}}$. In (d), we predefine a base ETF target and expand it for new tasks based on ProNC, and the degraded performance shows the benefit of naturally aligning the base ETF with feature learning in the first task. In (e), we replace the entire ProNC with a predefined fixed global ETF as in (Yang et al., 2023b) and the performance drops dramatically, corroborating the importance of ProNC for setting an appropriate ETF target in new task learning. In (f), we replace our cosine similarity based classifier by using the standard linear classifier, and the performance drop further highlights the usefulness of ETF in classification by providing equally separated feature representation targets for different classes.

*More comparison with contrastive learning based approaches.* Contrastive learning based approaches usually suffer from high computation costs due to the nature of contrastive learning with data augmentation. In the original implementation of these approaches, e.g., $Co^2L$ (Cha et al., 2021) and

Table 6: FAA, FF and total training time comparison with contrastive learning based approaches under their training setups.

| Buffer | Method | Seq-CIFAR-100 | | | Seq-TinyImageNet | | |
|---|---|---|---|---|---|---|---|
| | | Class-IL | Task-IL | Time(S) | Class-IL | Task-IL | Time(S) |
| 200 | $Co^2L$ (Cha et al., 2021) | 27.38 (67.82) | 42.37 (38.22) | 4362 | 13.88 (73.25) | 42.37 (47.11) | 12494 |
| | $MNC^3L$ (Dang et al., 2025) | 34.04 (52.40) | 59.46(33.66) | 3904 | 15.52 (52.07) | 44.59 (33.76) | 10922 |
| | **Ours** | **42.99 (36.07)** | **85.63 (4.40)** | 1482 | **27.44 (42.81)** | **69.09 (9.42)** | 12137 |
| 500 | $Co^2L$ (Cha et al., 2021) | 37.02 (51.23) | 62.44 (26.31) | 4380 | 20.12 (65.15) | 53.04 (39.22) | 12100 |
| | $MNC^3L$ (Dang et al., 2025) | 40.25 (46.09) | 65.85 (24.29) | 3979 | 20.31 (46.08) | 53.46 (26.45) | 12669 |
| | **Ours** | **48.94 (24.32)** | **86.38 (4.35)** | 1588 | **29.06 (38.58)** | **69.77 (9.52)** | 12350 |

$MNC^3L$ (Dang et al., 2025), 500 training epochs are used for the initial task, and followed by different epochs for each subsequent task, i.e., 100 epochs for Seq-CIFAR-100 and 50 epochs for Seq-TinyImageNet. To further demonstrate the superior performance of our approach, we evaluate our approach under the standard setup, against these approaches under their original implementation in Table 6. Clearly, while the performance of the contrastive learning based approaches improves with more training epochs, our approach still achieves significantly better performance with an even shorter training time.

*Hyperparameter robustness.* We conduct experiments to demonstrate the impact of $\lambda_1$ and $\lambda_2$ in Equation (5). As shown in Figure 3f, the performance of our approach is very stable under a wide range of $\lambda$ values, indicating the robustness of our approach.

*Time cost* To evaluate the computation efficiency of our approach, we summarize in Table 7 the average training time per epoch and per task for all approaches. It can be seen that our approach is more efficient than all considered replay-based approaches except iCaRL. iCaRL achieves higher efficiency by employing a binary cross-entropy loss, which enjoy a constant time complexity per class ($O(1)$), compared to the linear class-dependent complexity ($O(C)$, $C$ being the total number of classes) of conventional cross-entropy loss. Meanwhile, the per-epoch training time of our approach is comparable with the recent contrastive-learning based approaches. In summary, by

Table 7: Average training time per epoch and per task (seconds).

| Method | Epoch | Task |
|---|---|---|
| ER (Riemer et al., 2019) | 3.03 | 151.48 |
| iCaRL (Rebuffi et al., 2017) | 1.60 | 84.77 |
| GEM (Lopez-Paz & Ranzato, 2017) | 3.26 | 162.64 |
| GSS (Aljundi et al., 2019) | 28.86 | 1425.26 |
| DER (Buzzega et al., 2020) | 3.07 | 153.21 |
| DER++ (Buzzega et al., 2020) | 5.03 | 253.34 |
| $Co^2L$ (Cha et al., 2021) | 2.52 | 126.19 |
| CIIA (Wen et al., 2024) | 2.77 | 138.59 |
| $MNC^3L$ (Dang et al., 2025) | 2.82 | 141.01 |
| STAR (Eskandar et al., 2025) | 27.44 | 1371.89 |
| CSReL (Tong et al., 2025) | 8.42 | 433.35 |
| NCT (Yang et al., 2023b) | 3.41 | 172.02 |
| Ours | 2.91 | 146.71 |

leveraging NC in a principled way, our approach not only significantly outperforms baseline approaches by setting new SOTA performance, but also maintains a high efficiency due to the simplicity of our framework and robustness to buffer sizes. This highlights the great potential of our approach in practical resource-limited CL scenarios.

## 6 CONCLUSION

Neural collapse in DNN training characterizes an ideal feature learning target for CL with maximally and equally separated class prototypes through a simplex ETF, whereas recent studies leverage this by predefining a fixed global ETF target, suffering from impracticability and limited performance. To address this and unlock the potential of NC in CL, we propose a novel approach namely progressive neural collapse (ProNC), by initializing the ETF to align with first task leaning and progressively expanding the ETF for each new task without significantly shifting from the previous ETF. This will ensure maximal and equal separability across all encountered classes anytime during CL, without the global knowledge of total class numbers in CL. Building upon ProNC, we introduce a simple and flexible CL framework with minimal changes on standard CL frameworks, where the model is trained to push the learned sample features towards the corresponding ETF target and distillation with data replay is leveraged to further reduce forgetting. Extensive experiments have demonstrated the dominating performance of our approach against state-of-the-art baseline approaches, while maintaining superior efficiency and flexibility. One limitation here is that we assume clear task boundaries and it is interesting to see how our approach can be extended to more general setups. We hope this work will serve as initial steps for showcasing the great potentials of NC in facilitating better CL algorithm design and inspire further research in CL community along this interesting direction.

## REPRODUCIBILITY STATEMENT

In 4.1, explanations of implementation details for ProNC are presented. The details of hyperparameters, equipment, and code platform are presented in A.2. Furthermore, the code of our implementation is submitted together with the paper as supplement materials.

## ETHICS STATEMENT

We have read and followed the ICLR Code of Ethics.

## ACKNOWLEDGEMENTS

We want to thank all reviewers and area-chairs for their efforts toward improving this work.

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

APPENDIX

# A EXPERIMENT DETAILS

## A.1 DATASETS

**Seq CIFAR-10**: Based on CIFAR-10 dataset (Krizhevsky et al., 2009), this benchmark partitions 10 classes into 5 sequential tasks (2 classes per task), and each class has 5000 and 1000 $32 \times 32$ images each for training and testing, respectively;

**Seq CIFAR-100**: Constructed from CIFAR-100 (Krizhevsky et al., 2009), it comprises 10 tasks with 10 classes per task, and each class has 500 and 100 $32 \times 32$ images each for training and testing, respectively;

**Seq TinyImageNet**: Adapted from the TinyImageNet dataset (Le & Yang, 2015), this benchmark divides 200 classes into 10 sequential tasks (20 classes per task), and each class has 500 and 50 $64 \times 64$ images each for training and testing, respectively.

## A.2 IMPLEMENTATION DETAILS

All experiments are conducted on a single RTX 4090 GPU. For all datasets, we employ a modified ResNet18 network architecture (He et al., 2016), where the kernel size of the first convolutional layer is modified from 7×7 to 3×3, and the stride is changed from 2 to 1. The batch size is set to 32 across all experiments. The number of training epochs is set to 50 for Sequential CIFAR-10 and Sequential CIFAR-100, and 100 for Sequential TinyImageNet. For the buffer size, we use 200 and 500 in the main comparison table.

For ProNC, we consider the following hyperparameters: learning rate ($\eta$), weight of the alignment loss ($\lambda_1$), weight of the distillation loss ($\lambda_2$), momentum ($mom$), and weight decay ($wd$). The hyperparameters are selected through grid search. The chosen hyperparameters are presented in Table 8, and their corresponding search spaces are provided in Table 9.

Table 8: Hyperparameters of ProNC

| Method | Buffer size | Dataset | Hyperparameter |
|---|---|---|---|
| Ours | 200 | Seq-CIFAR-10 | $\eta$: 0.01, $\lambda_1$: 13, $\lambda_2$: 90, $mom$: 0, $wd$: 0 |
| | | Seq-CIFAR-100 | $\eta$: 0.03, $\lambda_1$: 18, $\lambda_2$: 170, $mom$: 0, $wd$: 0 |
| | | Seq-Tiny-ImageNet | $\eta$: 0.03, $\lambda_1$:20, $\lambda_2$: 165, $mom$: 0, $wd$: 0 |
| | 500 | Seq-CIFAR-10 | $\eta$: 0.01, $\lambda_1$:12 , $\lambda_2$: 80 , $mom$: 0, $wd$: 0 |
| | | Seq-CIFAR-100 | $\eta$: 0.03, $\lambda_1$:18 , $\lambda_2$: 170 , $mom$: 0, $wd$: 0 |
| | | Seq-Tiny-ImageNet | $\eta$: 0.03, $\lambda_1$:20 , $\lambda_2$: 200, $mom$: 0, $wd$: 0 |

Table 9: Search Spaces for Hyperparameters

| Hyperparameter | Values |
|---|---|
| $\eta$ | $\{0.01, 0.03, 0.05.0.1, 0.3\}$ |
| $\lambda_1$ | $\{5, 10, 12, 13, 18, 20\}$ |
| $\lambda_2$ | $\{50, 75, 80, 90, 165, 170, 200\}$ |
| $mom$ | $\{0, 0.9\}$ |
| $wd$ | $\{0, 10^{-5}, 5 \times 10^{-5}\}$ |

Our code is implemented based on the Continual Learning platform Mammoth (Boschini et al., 2022), which is also provided in the supplementary materials.

## B  MORE RESULTS

### B.1  FINAL AVERAGE ACCURACY WITH ERROR BARS

Table 10: Final average accuracies comparison under various setups. All results are averaged over multiple runs.

| Buffer | Method | Seq-Cifar-10 | | Seq-Cifar-100 | | Seq-Tiny-ImageNet | |
| --- | --- | --- | --- | --- | --- | --- | --- |
| | | Class-IL | Task-IL | Class-IL | Task-IL | Class-IL | Task-IL |
| 200 | ER  (Riemer et al., 2019) | $44.79 \pm 1.86$ | $91.19 \pm 0.94$ | $21.78 \pm 0.48$ | $60.19 \pm 1.01$ | $8.49 \pm 0.16$ | $38.17 \pm 2.00$ |
| | iCaRL  (Rebuffi et al., 2017) | $49.02 \pm 3.20$ | $88.99 \pm 2.13$ | $28 \pm 0.91$ | $51.43 \pm 1.47$ | $7.53 \pm 0.79$ | $28.19 \pm 1.47$ |
| | GEM  (Lopez-Paz & Ranzato, 2017) | $25.54 \pm 0.76$ | $90.44 \pm 0.94$ | $20.75 \pm 0.66$ | $58.84 \pm 1.00$ | – | – |
| | GSS  (Aljundi et al., 2019) | $39.07 \pm 5.59$ | $88.8 \pm 2.89$ | $19.42 \pm 0.29$ | $55.38 \pm 1.34$ | – | – |
| | DER  (Buzzega et al., 2020) | $61.93 \pm 1.79$ | $91.4 \pm 0.92$ | $31.23 \pm 1.38$ | $63.09 \pm 1.09$ | $11.87 \pm 0.78$ | $40.22 \pm 0.67$ |
| | DER++  (Buzzega et al., 2020) | $64.88 \pm 1.17$ | $91.92 \pm 0.60$ | $28.13 \pm 0.51$ | $66.80 \pm 0.41$ | $11.34 \pm 1.17$ | $43.06 \pm 1.16$ |
| | Co$^2$L  (Cha et al., 2021) | $51.27 \pm 1.86$ | $84.69 \pm 1.52$ | $18.09 \pm 0.49$ | $49.19 \pm 0.91$ | $12.95 \pm 0.06$ | $37.07 \pm 1.62$ |
| | CILA [47] | $59.68 \pm 0.65$ | $91.36 \pm 0.08$ | $19.49 \pm 0.53$ | $53.93 \pm 1.02$ | $12.98 \pm 0.10$ | $37.32 \pm 1.87$ |
| | MNC$^3$L  (Dang et al., 2025) | $52.20 \pm 1.56$ | $85.94 \pm 0.22$ | $15.81 \pm 0.48$ | $43.91 \pm 0.76$ | $10.57 \pm 1.66$ | $32.78 \pm 3.54$ |
| | STAR  (Eskandar et al., 2025) | $62.10 \pm 2.21$ | $93.54 \pm 1.48$ | $18.29 \pm 2.58$ | $58.53 \pm 13.06$ | $11.55 \pm 3.23$ | $41.70 \pm 0.95$ |
| | CSReL  (Tong et al., 2025) | $37.46 \pm 1.57$ | $69.22 \pm 3.03$ | $29.06 \pm 1.00$ | $66.99 \pm 0.35$ | $18.14 \pm 3.10$ | $45.04 \pm 5.86$ |
| | NCT  (Yang et al., 2023b) | $51.59 \pm 0.41$ | $80.63 \pm 0.46$ | $26.38 \pm 0.57$ | $75.75 \pm 0.17$ | $10.95 \pm 1.45$ | $52.71 \pm 4.12$ |
| | **Ours** | $\mathbf{65.58 \pm 0.15}$ | $\mathbf{96.86 \pm 0.10}$ | $\mathbf{42.99 \pm 0.85}$ | $\mathbf{85.63 \pm 0.73}$ | $\mathbf{27.44 \pm 1.00}$ | $\mathbf{69.09 \pm 0.65}$ |
| 500 | ER  (Riemer et al., 2019) | $57.74 \pm 2.48$ | $93.61 \pm 0.27$ | $22.35 \pm 0.61$ | $73.98 \pm 1.52$ | $9.99 \pm 0.29$ | $48.64 \pm 0.46$ |
| | iCaRL  (Rebuffi et al., 2017) | $47.55 \pm 3.95$ | $88.22 \pm 2.62$ | $33.25 \pm 1.25$ | $58.16 \pm 1.76$ | $9.38 \pm 1.53$ | $31.55 \pm 3.27$ |
| | GEM  (Lopez-Paz & Ranzato, 2017) | $26.2 \pm 1.26$ | $92.16 \pm 0.64$ | $25.54 \pm 0.65$ | $66.31 \pm 0.86$ | – | – |
| | GSS  (Aljundi et al., 2019) | $49.73 \pm 4.78$ | $91.02 \pm 1.57$ | $21.92 \pm 0.34$ | $60.28 \pm 1.18$ | – | – |
| | DER  (Buzzega et al., 2020) | $70.51 \pm 1.67$ | $93.40 \pm 0.21$ | $41.36 \pm 1.76$ | $71.73 \pm 0.74$ | $17.75 \pm 1.14$ | $51.78 \pm 0.88$ |
| | DER++  (Buzzega et al., 2020) | $72.70 \pm 1.36$ | $93.88 \pm 0.50$ | $38.20 \pm 1.00$ | $74.77 \pm 0.31$ | $19.38 \pm 1.41$ | $51.91 \pm 0.68$ |
| | Co$^2$L  (Cha et al., 2021) | $61.78 \pm 4.22$ | $89.51 \pm 2.45$ | $26.64 \pm 1.42$ | $62.32 \pm 0.19$ | $18.71 \pm 0.84$ | $50.74 \pm 1.24$ |
| | CILA  (Wen et al., 2024) | $67.82 \pm 0.33$ | $93.29 \pm 0.24$ | $31.27 \pm 0.17$ | $68.29 \pm 0.46$ | $18.09 \pm 0.49$ | $49.19 \pm 0.91$ |
| | MNC$^3$L  (Dang et al., 2025) | $52.20 \pm 1.56$ | $85.94 \pm 0.22$ | $22.29 \pm 0.18$ | $56.43 \pm 0.29$ | $11.52 \pm 0.01$ | $36.32 \pm 0.05$ |
| | STAR  (Eskandar et al., 2025) | $69.15 \pm 3.53$ | $95.36 \pm 0.37$ | $28.45 \pm 1.70$ | $74.06 \pm 1.63$ | $15.19 \pm 2.61$ | $55.06 \pm 2.07$ |
| | CSReL  (Tong et al., 2025) | $40.82 \pm 4.09$ | $73.30 \pm 5.92$ | $39.22 \pm 1.70$ | $75.09 \pm 0.78$ | $22.13 \pm 0.35$ | $51.94 \pm 0.22$ |
| | NCT (Yang et al., 2023b) | $60.93 \pm 0.94$ | $81.27 \pm 0.24$ | $33.84 \pm 0.38$ | $76.06 \pm 0.52$ | $18.24 \pm 0.62$ | $62.30 \pm 0.41$ |
| | **Ours** | $\mathbf{73.95 \pm 0.68}$ | $\mathbf{96.95 \pm 0.14}$ | $\mathbf{48.94 \pm 0.44}$ | $\mathbf{86.38 \pm 0.43}$ | $\mathbf{29.06 \pm 0.32}$ | $\mathbf{69.77 \pm 0.89}$ |

### B.2  FINAL FORGETTING WITH ERROR BARS

Table 11: Final forgetting comparison under various setups. All results are averaged over multiple runs.

| Buffer | Method | Seq-Cifar-10 | | Seq-Cifar-100 | | Seq-Tiny-ImageNet | |
| --- | --- | --- | --- | --- | --- | --- | --- |
| | | Class-IL | Task-IL | Class-IL | Task-IL | Class-IL | Task-IL |
| 200 | ER  (Riemer et al., 2019) | $59.30 \pm 2.48$ | $6.07 \pm 1.09$ | $75.06 \pm 0.63$ | $27.38 \pm 1.46$ | $76.53 \pm 0.51$ | $40.47 \pm 1.54$ |
| | iCaRL  (Rebuffi et al., 2017) | $23.52 \pm 1.27$ | $25.34 \pm 1.64$ | $47.20 \pm 1.23$ | $36.20 \pm 1.85$ | $31.06 \pm 1.91$ | $42.47 \pm 2.47$ |
| | GEM  (Lopez-Paz & Ranzato, 2017) | $80.36 \pm 5.25$ | $9.57 \pm 2.05$ | $77.40 \pm 1.09$ | $29.59 \pm 1.66$ | – | – |
| | GSS  (Aljundi et al., 2019) | $72.48 \pm 4.45$ | $8.49 \pm 2.05$ | $77.62 \pm 0.76$ | $32.81 \pm 1.75$ | – | – |
| | DER  (Buzzega et al., 2020) | $35.79 \pm 2.59$ | $6.08 \pm 0.70$ | $62.72 \pm 2.69$ | $25.98 \pm 1.55$ | $64.83 \pm 1.48$ | $40.43 \pm 1.05$ |
| | DER++  (Buzzega et al., 2020) | $32.59 \pm 2.32$ | $5.16 \pm 0.21$ | $60.99 \pm 1.52$ | $23.91 \pm 0.55$ | $73.47 \pm 1.23$ | $39.02 \pm 0.43$ |
| | Co$^2$L  (Cha et al., 2021) | $30.17 \pm 8.57$ | $2.91 \pm 5.25$ | $64.14 \pm 0.69$ | $36.81 \pm 0.63$ | $62.04 \pm 0.28$ | $40.75 \pm 3.55$ |
| | CILA  (Wen et al., 2024) | $37.52 \pm 5.84$ | $5.49 \pm 0.74$ | $64.01 \pm 0.18$ | $33.07 \pm 0.96$ | $63.11 \pm 0.61$ | $41.40 \pm 0.51$ |
| | MNC$^3$L  (Dang et al., 2025) | $33.74 \pm 1.65$ | $4.90 \pm 0.15$ | $62.51 \pm 0.40$ | $39.79 \pm 0.98$ | $59.68 \pm 2.02$ | $45.10 \pm 1.57$ |
| | STAR  (Eskandar et al., 2025) | $21.78 \pm 3.16$ | $6.23 \pm 2.06$ | $68.40 \pm 8.52$ | $27.84 \pm 8.01$ | $67.43 \pm 1.48$ | $34.78 \pm 2.08$ |
| | CSReL  (Tong et al., 2025) | $26.34 \pm 2.18$ | $17.16 \pm 2.91$ | $58.23 \pm 1.54$ | $23.20 \pm 1.74$ | $49.77 \pm 2.27$ | $34.12 \pm 2.18$ |
| | NCT  (Yang et al., 2023b) | $\mathbf{22.48 \pm 19.50}$ | $1.41 \pm 1.09$ | $\mathbf{27.40 \pm 1.65}$ | $4.79 \pm 0.07$ | $49.33 \pm 4.47$ | $15.88 \pm 0.95$ |
| | **Ours** | $32.75 \pm 4.71$ | $\mathbf{0.65 \pm 0.08}$ | $36.07 \pm 0.51$ | $\mathbf{4.40 \pm 0.82}$ | $\mathbf{42.81 \pm 0.56}$ | $\mathbf{9.42 \pm 0.58}$ |
| 500 | ER  (Riemer et al., 2019) | $43.22 \pm 2.10$ | $3.50 \pm 0.53$ | $73.08 \pm 0.78$ | $16.23 \pm 1.06$ | $75.21 \pm 0.54$ | $30.73 \pm 0.62$ |
| | iCaRL  (Rebuffi et al., 2017) | $28.20 \pm 2.41$ | $22.61 \pm 3.97$ | $40.99 \pm 1.02$ | $27.90 \pm 1.37$ | $37.30 \pm 1.42$ | $39.44 \pm 0.84$ |
| | GEM  (Lopez-Paz & Ranzato, 2017) | $78.93 \pm 6.53$ | $5.60 \pm 0.96$ | $71.34 \pm 0.78$ | $20.44 \pm 1.13$ | – | – |
| | GSS  (Aljundi et al., 2019) | $59.18 \pm 4.00$ | $6.37 \pm 1.55$ | $74.12 \pm 0.42$ | $26.57 \pm 1.34$ | – | – |
| | DER  (Buzzega et al., 2020) | $24.02 \pm 1.63$ | $3.72 \pm 0.55$ | $49.07 \pm 2.54$ | $25.98 \pm 1.55$ | $59.95 \pm 2.31$ | $28.21 \pm 0.97$ |
| | DER++  (Buzzega et al., 2020) | $22.38 \pm 4.41$ | $4.66 \pm 1.15$ | $49.18 \pm 2.19$ | $15.75 \pm 0.48$ | $58.75 \pm 1.93$ | $25.47 \pm 1.03$ |
| | Co$^2$L  (Cha et al., 2021) | $17.79 \pm 3.36$ | $2.65 \pm 1.00$ | $48.60 \pm 1.34$ | $23.47 \pm 0.27$ | $49.64 \pm 1.14$ | $13.80 \pm 2.10$ |
| | CILA  (Wen et al., 2024) | $18.22 \pm 1.32$ | $0.65 \pm 0.13$ | $45.67 \pm 1.02$ | $16.98 \pm 0.56$ | $64.14 \pm 0.69$ | $27.83 \pm 1.32$ |
| | MNC$^3$L  (Dang et al., 2025) | $27.88 \pm 2.75$ | $3.15 \pm 0.42$ | $46.09 \pm 0.58$ | $24.29 \pm 0.50$ | $44.96 \pm 0.09$ | $39.00 \pm 0.17$ |
| | STAR  (Eskandar et al., 2025) | $18.59 \pm 2.91$ | $2.38 \pm 0.34$ | $53.11 \pm 1.57$ | $15.22 \pm 0.96$ | $63.42 \pm 3.19$ | $15.19 \pm 2.61$ |
| | CSReL  (Tong et al., 2025) | $29.18 \pm 6.91$ | $16.89 \pm 3.19$ | $47.03 \pm 2.57$ | $15.55 \pm 0.76$ | $50.01 \pm 1.87$ | $28.20 \pm 0.36$ |
| | NCT (Yang et al., 2023b) | $\mathbf{13.82 \pm 3.64}$ | $3.84 \pm 0.35$ | $24.37 \pm 5.42$ | $\mathbf{3.97 \pm 0.75}$ | $50.90 \pm 1.16$ | $\mathbf{6.84 \pm 0.87}$ |
| | **Ours** | $26.75 \pm 1.29$ | $\mathbf{0.24 \pm 0.14}$ | $\mathbf{24.32 \pm 0.24}$ | $4.35 \pm 0.17$ | $\mathbf{38.58 \pm 0.39}$ | $9.52 \pm 0.96$ |

## C  PROOF OF THEOREM 1

Based on the definition of ETF, we know that $\mathbf{E}^*$ can be expressed as

$$\mathbf{E}^* = \sqrt{\frac{K_1}{K_1 - 1}} \mathbf{U}^* \left( \mathbf{I}_{K_1} - \frac{1}{K_1} \mathbf{1}_{K_1} \mathbf{1}_{K_1}^\top \right), \tag{6}$$

where $\mathbf{U}^* \in \mathbb{R}^{d \times K_1}$ denotes the corresponding orthogonal matrix.

Because $\mathbf{U}' = \sqrt{\frac{K_1 - 1}{K_1}} \tilde{M}_{K_1} \left( \mathbf{I}_{K_1} - \frac{1}{K_1} \mathbf{1}_{K_1} \mathbf{1}_{K_1}^\top \right)$, we can have

$$\tilde{M} = \sqrt{\frac{K_1}{K_1 - 1}} \mathbf{U}' \left( \mathbf{I}_{K_1} - \frac{1}{K_1} \mathbf{1}_{K_1} \mathbf{1}_{K_1}^\top \right). \tag{7}$$

This is true since $\mathbf{I}_{K_1} - \frac{1}{K_1} \mathbf{1}_{K_1} \mathbf{1}_{K_1}^\top$ is an orthogonal projection matrix with rank $K_1 - 1$ and also equal to its pseudoinverse.

It is clear that

$$\tilde{M}_{K_1} - \mathbf{E}^* = \sqrt{\frac{K_1}{K_1 - 1}} \mathbf{U}' \left( \mathbf{I}_{K_1} - \frac{1}{K_1} \mathbf{1}_{K_1} \mathbf{1}_{K_1}^\top \right) - \sqrt{\frac{K_1}{K_1 - 1}} \mathbf{U}^* \left( \mathbf{I}_{K_1} - \frac{1}{K_1} \mathbf{1}_{K_1} \mathbf{1}_{K_1}^\top \right)$$

$$= \sqrt{\frac{K_1}{K_1 - 1}} \left( \mathbf{U}' - \mathbf{U}^* \right) \left( \mathbf{I}_{K_1} - \frac{1}{K_1} \mathbf{1}_{K_1} \mathbf{1}_{K_1}^\top \right).$$

Since $\sqrt{\frac{K_1}{K_1 - 1}}$ is a scalar and $\mathbf{I}_{K_1} - \frac{1}{K_1} \mathbf{1}_{K_1} \mathbf{1}_{K_1}^\top$ is a fixed projection matrix, finding $\mathbf{E}^*$ to minimize $\|\tilde{M}_{K_1} - \mathbf{E}^*\|_F^2$ is equivalent to finding an orthogonal matrix $\mathbf{U}^*$ that minimizes $\|\mathbf{U}' - \mathbf{U}^*\|_F^2$.

To this end, the following lemma (Zou et al., 2006) characterizes the nearest orthogonal matrix to any real matrix.

**Lemma 1** (Nearest Orthogonal Matrix via SVD). *Let $\mathbf{A} \in \mathbb{R}^{m \times n}$ be a real matrix with Singular Value Decomposition (SVD) $\mathbf{A} = \mathbf{W} \boldsymbol{\Sigma} \mathbf{V}^\top$, where $\mathbf{W} \in \mathbb{R}^{m \times s}$ and $\mathbf{V} \in \mathbb{R}^{s \times n}$ are orthogonal matrices, and $\boldsymbol{\Sigma} \in \mathbb{R}^{s \times s}$ is a diagonal matrix of singular values with $s = \min(m, n)$. The nearest orthogonal matrix $\mathbf{Q} \in \mathbb{R}^{m \times n}$ to $\mathbf{A}$ under Frobenius norm is:*

$$\mathbf{Q} = \mathbf{W} \mathbf{V}^\top.$$

Therefore, given the SVD of $\mathbf{U}'$ as $\mathbf{W} \boldsymbol{\Sigma} \mathbf{V}^\top$, the orthogonal matrix closest to $\mathbf{U}'$ can be represented as $\mathbf{U}^* = \mathbf{W} \mathbf{V}^\top$, which will lead to an ETF matrix

$$\mathbf{E}^* = \sqrt{\frac{K_1}{K_1 - 1}} \mathbf{W} \mathbf{V}^\top \left( \mathbf{I}_{K_1} - \frac{1}{K_1} \mathbf{1}_{K_1} \mathbf{1}_{K_1}^\top \right).$$

## D   LLM USAGE

During this project, we did not use LLM.

