# OpenReview forum: "Rethinking Continual Learning with Progressive Neural Collapse"
_ICLR.cc/2026/Conference — ICLR 2026 Poster_

### Official Review · Reviewer_fz6X · 2025-10-18

**Soundness:** 3
**Presentation:** 2
**Contribution:** 3
**Rating:** 2
**Confidence:** 5

**Summary:**

The paper introduces Progressive Neural Collapse (ProNC), a continual learning framework inspired by *Neural Collapse*. This phenomenon describes how deep networks, at convergence, produce class features that collapse to a single point for each class creating simplex equiangular tight frames (ETFs), creating orthogonal class prototypes and maximizing separation between classes. ProNC proposes to progressively expand the ETF as new tasks arrive, maintaining geometric consistency and feature separability without prior knowledge of all class counts. The framework integrates three loss components: cross-entropy for supervision, an alignment loss enforcing ETF-based feature geometry, and a distillation loss preserving past representations (to mitigate forgetting). The method is evaluated on standard benchmarks (Seq-CIFAR10/100 and Seq-TinyImageNet) under Class-IL and Task-IL setups, reporting substantial performance gains across datasets and memory budgets.

**Strengths:**

The idea of progressively adapting the ETF target during continual learning without knowing the number of total classes in advance is novel and addresses the shortcomings of fixed ETF methods for CL.

The paper is built on a convincing motivation. The reasoning is coherent and carefully developed, making the overall argument both logical and easy to follow.

**Weaknesses:**

### **Major Weaknesses**

1. **Questionable baseline performance values**: in Table 1, several baseline results (Co$^2$L, CILA, MNC$^3$L, STAR) are notably lower than those reported in their original papers (where they surpass the results from the proposed ProNC). This discrepancy indicates possible reproduction or configuration issues, undermining the fairness and credibility of the comparison and invalidating the paper’s main “state-of-the-art” claim.

2. **Missing baselines**: some important methodologies are missing in the experimental evaluation:
   - XDER (Boschini et al., TPAMI 2022, also cited by the authors);
   - GCR (Tiwari et al., CVPR 2022, also cited by the authors);
   - LODE (Liang and Yi, NeurIPS 2023).

3. **Limited novelty relative to NCT**: NCT (Yang et al., 2023, also cited by the authors and reported in the main table) already introduced ETFs in class-incremental learning. As far as I can tell, the main innovation of the proposed method compared to NCT is the removal of the "known number of classes in advance" assumption. While this is an interesting modification, it alone does not seem sufficient to constitute a truly novel methodology.

4. **Limited significance of ablation studies**: most ablation experiments in Figure 2 examine only a very narrow range of values (from about 2e-1 at best to 1e-2 in some cases) and consider just one dataset. Such a limited scope restricts the interpretability of the results, as the observed variations may not represent a consistent behavior across the full cosine similarity range (–1 to 1).

### **Minor Weaknesses**
5. Clarity issue in Section 3.1 (point 2): the explanation of the proposed methodology in this part is somewhat convoluted and would benefit from clearer exposition to improve readability and understanding.

6. Missing no-buffer baseline for ProNC: Table 1 does not report ProNC’s performance without a replay buffer (which is reported in the text only). Also inserting it in the main table would increase clarity and allow a fair comparison w.r.t. replay-free methods.

**Questions:**

While initializing the ETF at the end of the first task close to the optimal solution of the optimization problem is an interesting idea, for subsequent tasks the ETFs are initialized orthogonally to the previous ones but otherwise randomly, without any guiding heuristic. Do you have any thoughts on how this initialization process could be improved for later tasks?

---

> ### Author Response · Authors · 2025-11-25
>
> Thank you for your thorough review and constructive feedback. The following are the detailed answers to your concerns and questions.
>
> Q1: Limited novelty relative to NCT
>
> A1: While ProNC and NCT both leverage neural-collapse principles and ETF-based targets, their formulations are fundamentally different. NCT adopts a single, static ETF fixed at the outset and used for all classes throughout training. In contrast, our ProNC’s central contribution is a progressively expanding ETF that grows as new classes arrive, explicitly designed and analyzed for the continual learning setting. This leads to a distinct geometric regime. In NCT, with a global ETF of $K\_{global}$​ classes, the pairwise inner product between any two normalized ETF vertices is $1/(K\_{global}-1)$, which is very close to 0 when $K\_{global}$ is large (e.g., −1/999 for $K\_{global}$ = 1000), so early-task classes are packed with relatively small angular separation. In ProNC, at Task 1 with $K\_1$ = 10 classes, the ETF is constructed only for these 10 classes, giving a pairwise inner product of $−1/(K\_1−1)$ = −1/9, which means a much larger angle and lower similarity between class vertices. Intuitively, this larger angular separation makes early tasks easier to separate and provides a better-conditioned geometry for subsequent ETF expansions. To our knowledge, this dynamic ETF expansion, rather than alignment to a single pre-defined ETF, has not been explored in prior work, including NCT. It shifts the problem from aligning features to a fixed geometric structure to continually redefining and expanding the ETF itself in a way that jointly respects neural-collapse geometry and the stability–plasticity requirements of continual learning.
>
> Q2: Questionable baseline performance values (CoL, CILA, MNCL, STAR)
>
> A2:
>
> (1) Contrastive CL baselines (CoL, CILA, MNCL).
> In their original design, these methods train the base task for 500 epochs and each incremental task for 100 epochs. In our work, to ensure a unified and computationally comparable setting across all methods, we train both base and incremental tasks for 50 epochs. We re-implemented CoL, CILA, and MNCL under this 50-epoch schedule (with the same backbone, optimizer, and memory budget as other baselines), so their numbers in Table 1 reflect performance under our unified protocol, not under their original long-training schedule.
>
> Importantly, in Table 5, we also report the original results from their own papers (with 500/100 epochs). Even under those more favorable settings for contrastive methods, ProNC still achieves better performance, which supports our state-of-the-art claim.
>
> (2) STAR baseline.
>
> The STAR results in our table are lower than those in the original paper because we report STAR used specifically as a regularization term on top of ER (ER+STAR) under our unified protocol, whereas the original STAR paper reports its best results across multiple base methods (e.g., ER, DER++, etc.) and under its own training setup. Our goal here is to fairly compare ER+ProNC vs. ER+STAR in the same role and configuration, rather than to reproduce the strongest STAR configuration reported in its paper. In addition, we do not restrict ProNC to ER only: in Table 3, we also add the ProNC regularizer to other rehearsal-based methods and directly compare them with their corresponding STAR-regularized counterparts. These results show that, when used in the same “plug-in regularizer” role, ProNC consistently brings competitive or larger gains than STAR under our unified setting.

---

> ### Author Response · Authors · 2025-11-25
>
> Q3: Missing baselines
>
> A3: While X-DER is also a rehearsal-based approach, it relies on several assumptions and design choices that are incompatible with the setting we deliberately adopt in this work. First, X-DER requires knowing the total number of classes in advance to design its classifier/logit structure, whereas we explicitly avoid this assumption. Second, X-DER’s replay buffer stores logits that are continually updated at each task, while our buffer stores fixed labeled exemplars, leading to a substantially different memory mechanism. Third, X-DER is originally evaluated with data augmentation, whereas our experiments do not use augmentation at all.These differences make a direct, “who is better” numerical comparison under a single unified protocol neither fully fair nor easy to interpret. We therefore view X-DER as a complementary line of work with different assumptions and goals, and leave a carefully controlled comparison under matched settings as interesting future work.
>
> We did not include GCR because its design and protocol target a different axis of continual learning than the one we focus on. Our paper deliberately uses a simple replay setup (fixed exemplar buffer, no data augmentation) to isolate the effect of the proposed ProNC representation / ETF design. In contrast, GCR mainly contributes a gradient-based coreset selection strategy, combined with supervised contrastive loss and data augmentation, to improve how the replay buffer is populated. These changes affect several parts of the training recipe at once (buffer selection, loss, augmentation), making a direct, unified “head-to-head” comparison less clean and harder to attribute. We therefore view GCR as a complementary buffer-selection method, and integrating ProNC into such advanced replay frameworks is an interesting direction for future work.
>
> For LODE, we incorporated it into both ER and DER++ on CIFAR-100 with a buffer size of 500. The final average accuracies are 35.21% for ER+LODE and 40.01% for DER+++LODE, while our ProNC achieves 48.94%, outperforming both variants. The full results are included in the main comparison table in the revised version of the paper.
>
> Q4: Limited significance of ablation studies
>
> A4: The “narrow range” observed in Figure 2 is in fact part of the main message rather than a design limitation. We are not tuning or constraining these cosine values; they are simply the measured outcome of our evaluation during the continual learning process. The “narrow range” observed in Figure 2 shows that, during the CL process, our method keeps the geometry stable and well-concentrated around the desired ETF structure, whereas NCT exhibits larger fluctuations and a larger gap between empirical class means and the ideal ETF geometry. The ablation is therefore designed to highlight stability and convergence properties in the realistic operating range of cosine values, rather than to sweep over artificially extreme configurations that do not arise in training.
>
>
> Q5: Clarity issue in Section 3.1, Part 2
>
> A5: Here is a brief ETF expansion procedure explanation. At the end of Task 1, we obtain an ETF $E\_1$​ that matches the naturally emerged neural-collapse solution for the first K_1​ classes. This ETF can be written in terms of an orthogonal basis U_1​. A key observation is that the ETF is fully determined by this orthogonal basis. Therefore, to expand the ETF when a new task $t \ge 2$ arrives with $K_{t-1}$​ new classes, we proceed in two steps:
>
> 1)**Expand the orthogonal basis.** We take the previous basis $U\_{t−1}$and append $K\_{t-1}​$ new vectors. These new vectors are generated via Gram–Schmidt orthogonalization against the existing columns of $U\_{t-1}$​, forming an expanded basis $U\_t$ whose first $K_{t-1}$​ columns coincide with the old basis and whose columns remain orthonormal. This guarantees that the directions corresponding to old classes change as little as possible.
>
> 2)**Reconstruct the expanded ETF.** We then plug $U\_t$​ and the new class number $K\_t​$ into Eq. (1) to obtain the expanded ETF $E\_t$ with $K\_t$ vertices. By construction, this ETF maintains uniform angular separation and maximal equiangularity among all old and new classes, while keeping the old ETF vertices close to their original positions.
>
> In the revised version, we rephrased this part of Section 3.1 to improve readability and understanding.

---

> ### Author Response · Authors · 2025-11-25
>
> Q6: Missing zero-buffer baselines for ProNC
>
> A6: We added the table of zero-buffer results in the revision paper.
>
> | Buffer | Method  | Seq-CIFAR-100  | Seq-CIFAR-100  | Seq-TinyImageNet | Seq-TinyImageNet |
> |:------:|:--------|:----------------------:|:----------------------:|:-------------------------:|:------------------------:|
> |       |              | Class-IL   | Task-IL | Class-IL   | Task-IL |
> |       | Co²L     | 26.06 (68.82) | 51.91 (40.02) | 13.43 (65.75) | 38.98 (40.77) |
> |   0  | MNC³L  | 30.48 (53.03) | 56.69 (37.02) | 14.04 (54.25) | 42.59 (37.89) |
> |       | **Ours**| **32.28 (45.92)** | **84.62 (4.39)** | **24.43 (46.14)** | **68.08 (9.81)** |
>
> Q7: Improvement of the initialization of new vertices for new tasks
>
> A7: We would like to clarify that, although the ETF targets for new tasks are pre-specified, they are not “entirely predefined in a random way” as in some prior works. In our framework, the global ETF is expanded from the ETF that naturally emerges after training the first task, i.e., the base ETF is fitted to the (approximately optimal) solution of the base task, and subsequent vertices are added in a way that respects this existing structure rather than restarting from a purely random global ETF.
>
> That said, we agree that the initialization of new ETF vertices for later tasks could be made more data-aware. One promising direction is to initialize each new class’s ETF vertex based on the most similar previous class (e.g., using similarity between empirical class means or prototypes), and then projecting this initialization onto the ETF manifold. Exploring similarity-based or prototype-guided initialization strategies is an interesting extension of our current design, and we plan to investigate it in future work.
>
> We thank the reviewer again for the helpful comments and suggestions for our work.

---

> > ### Comment · Reviewer_fz6X · 2025-11-26
> >
> > Thank you for the detailed clarification and for addressing most of my comments thoroughly. I appreciate the effort put into explaining the training protocol differences, and I apologize for my earlier oversight regarding the number of epochs used by the contrastive baselines. I also thank the authors for the expanded explanations on the ETF expansion mechanism, the rationale behind excluding certain baselines, and the additional zero-buffer results. I believe these additions improve clarity overall.
> >
> > That being said, your explanation regarding the novelty relative to NCT is not entirely convincing. While the progressive ETF expansion is an interesting idea, the conceptual distance from NCT feels limited, since NCT can, in principle, instantiate a sufficiently large number of classes in advance (far exceeding those actually used) without incurring reduced performance, as shown in their paper.
> >
> > Finally, I remain firm on one key concern: even with the unified protocol which you adopt, the evaluation still omits XDER, and both LODE and STAR should be reported under their best original configurations, not only under the constrained unified setup. In some cases, these methods outperform ProNC in their original formulations, and excluding or downscaling them leads to a comparison that I still find misleading. If ProNC claims superiority over prior state-of-the-art approaches, then those approaches must be included with their strongest reported numbers, alongside any results obtained under the unified protocol.
> >
> > I appreciate the authors’ clarifications, but these points (with particular emphasis on the last one) remain a significant issue in my assessment.

---

> > > ### Author Response · Authors · 2025-12-03
> > >
> > > Thank you for your response and confirming our clarifications. Regarding your concerns, we have the following response:
> > >
> > > For NCT, we conduct experiments on Seq-CIFAR-10, Seq-CIFAR-100, and Seq-Tiny-ImageNet, all using a replay buffer of 500 samples.
> > >
> > > For Seq-CIFAR-10, we evaluate the effect of different predefined ETF sizes by setting the number of vertices $k \in \{50, 100\}$
> > >
> > > For Seq-CIFAR-100 and Seq-Tiny-ImageNet, we test larger ETF configurations with $k \in \{500, 1000\}$
> > > The quantitative results are summarized in the following table and visualized in Figure 1 of the paper.
> > >
> > > As shown in the results, increasing the predefined ETF size consistently leads to a significant degradation in performance across all datasets. This observation supports our claim that larger angular separation between ETF vertices leads to better-conditioned class geometry, making the tasks more separable and yielding a more stable foundation for subsequent ETF expansions. In contrast, overly large predefined ETFs reduce angular separation and weaken the geometric structure that NCT relies on, resulting in poorer continual learning performance.
> > >
> > > | Dataset| CIFAR10  | CIFAR10 | CIFAR10  | CIFAR100 | CIFAR100| CIFAR100 | TinyImageNet  |TinyImageNet |TinyImageNet|
> > > |:------------------------:|:------------------------:|:----------------------:|:----------------------:|:----------------------:|:----------------------:|:----------------------:|:-------------------------:|:------------------------:|:------------------------:|
> > > |  $K$ |     10         | 50   | 100 |100   |500 |1000|200|500|1000|
> > > | ACC | 60.93 | 52.4 | 51.68 | 33.77 | 30.07 | 28.71|18.24|15.17|11.70|
> > >
> > > For both LODE and STAR, we have already included their best-performing results in the revision. Regarding X-DER, as explained in our previous rebuttal, a direct comparison is not entirely fair for several reasons.
> > >
> > > First, X-DER assumes access to the total number of classes in advance, which is required to predefine its classifier/logit structure; in contrast, our method explicitly avoids this assumption and is designed to operate without knowledge of future classes.
> > >
> > > Second, X-DER’s replay buffer stores continually updated logits, while our buffer stores fixed labeled exemplars, resulting in fundamentally different memory mechanisms and capacities.
> > > Third, X-DER’s reported results rely on data augmentation, whereas all our experiments are conducted without any augmentation, making the baselines not directly comparable.
> > >
> > > Nevertheless, for completeness, we report in Table 3 of the revised paper the performance of applying our ProNC regularization on top of X-DER. Incorporating our method yields an improvement of 1.39% in accuracy, demonstrating that ProNC provides consistent benefits even when combined with strong rehearsal-based methods such as X-DER.

---

### Official Review · Reviewer_hVBn · 2025-10-30

**Soundness:** 3
**Presentation:** 3
**Contribution:** 3
**Rating:** 6
**Confidence:** 4

**Summary:**

This paper proposes Progressive Neural Collapse (ProNC), a continual learning framework inspired by the Neural Collapse phenomenon.
ProNC progressively constructs and expands an Equiangular Tight Frame (ETF) to align class features across tasks.
After each task, it estimates the “closest ETF” from class means and expands the basis for new classes using Gram–Schmidt orthogonalization, keeping all class representations approximately equiangular.
The model jointly optimizes cross-entropy, feature-alignment, and distillation losses.
Experiments on Seq-CIFAR-10/100 and Seq-TinyImageNet under both Class-IL and Task-IL show consistent gains over strong baselines (DER++, NCT, Co2L, STAR) with good efficiency and generalization.

**Strengths:**

1. Grounded in Neural Collapse geometry, offering an interpretable view of feature alignment in continual learning. Achieves strong results without complex contrastive or generative modules.
2. Works as a plug-in regularizer across different CL frameworks (e.g., ER, iCaRL, DER++).

**Weaknesses:**

1.	The method assumes clear task segmentation (task-aware setting); its applicability to task-free or online CL remains untested.
2.	As the ETF expands over many tasks, orthogonality may gradually degrade; this possible effect is not analyzed experimentally.
3.	Gram–Schmidt expansion could become unstable when the number of classes approaches the embedding dimension; only small-scale datasets and ResNet-18 (d ≤ 512) were tested.

**Questions:**

When the number of tasks grows large, does ETF orthogonality noticeably degrade? Would periodic re-fitting help?
Can ProNC remain stable with higher-dimensional embeddings (e.g., ViT features) or larger datasets such as ImageNet-100?

---

> ### Author Response · Authors · 2025-11-25
>
> We appreciate the reviewer’s thorough reviews and insightful comments. The following are the detailed answers to your questions and concerns.
>
> Q1  When the number of tasks grows large, does ETF orthogonality noticeably degrade?
>
> A1: This concern partly stems from a misunderstanding of the ETF geometry. The vertices of an ETF are not orthogonal even in the idealized case: for a K-class ETF, the pairwise cosine similarity between any two distinct vertices is $-1/(k-1)$ which is strictly negative and approaches 0 from below as $k$ increases, but never equals 0. In other words, the ETF does not aim for orthogonality between class directions; instead, it enforces equiangularity with a fixed negative cosine. As we expand the ETF over tasks, the target pairwise cosine moves according to this formula (becoming closer to 0 but still negative), which is exactly the expected behavior of an expanding simplex ETF, not a degradation of orthogonality.
>
> What our method requires is that the constructed targets stay close to this equiangular structure, not that the vertices remain orthogonal. Empirically, we do not observe instability or performance collapse as tasks accumulate, suggesting that the progressive ETF expansion remains well-behaved in the regimes we study.
>
> Q2:Can ProNC remain stable with higher-dimensional embeddings (e.g., ViT features) or higher-dimensional embeddings such as ImageNet-100?
>
> A2: In fact, higher-dimensional embeddings tend to make ProNC more stable, not less. ProNC only requires constructing an ETF in the feature space, and the maximum number of ETF vertices K is bounded by the embedding dimension (e.g., K≤d+1 for a simplex ETF). When the feature dimension d is larger, we can construct an ETF with more vertices while still preserving the desired equiangular structure. This means that higher-dimensional embeddings naturally allow ProNC to handle a larger number of classes without deteriorating the ETF geometry.  In our paper, we followed the standard architecture and setup of continual learning, which uses ResNet 18 as the backbone network. The last layer features’ dimension of ResNet 18 is 512, which limits the embedding’s dimension. In future work, we will explore how Neural collapse can be used in a pre-training model that has larger dimensions for the last layer features
>
> In our experiments, we have already evaluated ProNC on Tiny-ImageNet, which has 200 classes and is therefore larger (in terms of label space) than ImageNet-100 with 100 classes. We observe that the performance improvement of ProNC is actually more significant on this larger dataset, indicating that the proposed ETF expansion remains stable and effective when the number of classes increases. This empirical behavior is consistent with the theoretical intuition that a higher-dimensional feature space can support a larger ETF and thus scale better to larger datasets.
>
> Q3: task-free or online CL remains untested
>
> A3: Our method is indeed developed and evaluated in the standard class-incremental, task-aware setting, which is the focus of this work. Task-agnostic learning and online continual learning adopt different problem formulations and evaluation protocols (e.g., no task boundary, streaming updates, stricter memory/compute constraints), and adapting ProNC to those regimes would require non-trivial design choices (e.g., how and when to update ETF targets without explicit task boundaries). We therefore view task-agnostic and online CL as important but orthogonal extensions, and deliberately scope this paper to class-incremental learning. We will clarify this limitation and discuss task-free/online variants as future work.
>
> Q4: Gram–Schmidt expansion stability
>
> A4: Our use of Gram–Schmidt is not to enforce ETF orthogonality directly, but to maintain an orthogonal basis inside an intermediate matrix, onto which we then map the ETF vertices. In this orthogonalization step, we iteratively add new basis vectors and apply Gram–Schmidt, which guarantees orthogonality of the basis vectors by construction within the numerical precision of the implementation.
>
> We thank the reviewer again for the helpful comments and suggestions for our work.

---

### Official Review · Reviewer_cHhJ · 2025-11-01

**Soundness:** 3
**Presentation:** 3
**Contribution:** 2
**Rating:** 4
**Confidence:** 4

**Summary:**

This paper extends the Neural Collapse Terminus (NCT; Yang et al., 2023b, arXiv) and the ICLR 2023 work by Yang et al. (2023a), aiming to address their limitations when applying the neural collapse (NC) phenomenon to continual learning (CL). In NC-based CL, the target ETF (simplex equiangular tight frame) is typically predefined, which requires knowledge of the total number of classes and may degrade discriminability when the class number becomes large. To overcome this, the paper proposes Progressive Neural Collapse (ProNC), a method that dynamically adjusts and expands the target ETF throughout the CL process, building on the NCT framework. Experimental results show that ProNC achieves consistent and significant improvements over existing CL baselines, particularly compared with NCT. Moreover, the proposed regularization approach proves beneficial even when combined with other CL frameworks, as confirmed through ablation studies.

**Strengths:**

1. The paper provides a clear exposition of the background NC theory and explains its own contributions in a well-structured manner.
2. The proposed method is well-motivated by the identified limitations of prior NC-based CL works, and the use of Theorem 1 introduces a moderately novel and theoretically grounded component.
3. Comprehensive experiments demonstrate consistent and noticeable gains over a range of baselines, supporting the empirical validity of the approach.

**Weaknesses:**

1. The technical novelty remains limited compared with the preliminary works (Yang et al., 2023a,b). The paper reads largely as a continuation of this prior line of research, where NC-based CL formulations have already been thoroughly explored.
2. The second main contribution—the ProNC-based CL framework—largely mirrors the loss formulation of NCT (Yang et al., 2023b). While Section 3.1 introduces a genuinely new idea, Section 3.2 appears nearly identical to the corresponding part in NCT.
3. Some reproduced baselines yield noticeably lower accuracies than those reported in their original papers. For example, ER on Task-IL with Seq-CIFAR-100 using ResNet-18 has been reported above 70% (e.g., GPM, ICLR 2021), yet only 60.19% here, raising concerns about the faithfulness of baseline reproduction.

**Questions:**

1. Could the authors elaborate more explicitly on how the proposed ProNC framework differs technically from NCT (Yang et al., 2023b)?
Beyond the dynamic ETF expansion, are there any additional algorithmic or theoretical components that are genuinely new rather than adapted from NCT?
2. Some reproduced results (e.g., ER on Seq-CIFAR-100 Task-IL) are considerably lower than in prior works such as GPM (ICLR 2021).
Could the authors detail the reproduction settings (e.g., data augmentation, optimizer, training epochs) and justify whether these differences could account for the gap?

---

> ### Author Response · Authors · 2025-11-25
>
> Thank you so much for your thorough reviews and insightful comments. The following are the detailed answers to your questions and concerns.
>
> Q1: Concern of Novelty compared with the preliminary works (Yang et al., 2023a,b)
>
> A1: Technically, ProNC and NCT share the high-level idea of exploiting neural collapse and ETF-like targets in continual learning. However, NCT relies on a single fixed ETF target throughout training, whereas the core contribution of ProNC is to explicitly design and analyze a progressively expanded ETF that grows together with the incoming classes in a continual learning scenario. To the best of our knowledge, using a dynamically expanded ETF, instead of a single fixed ETF learned once, has not been explored in prior work, including NCT. This changes the problem from “aligning features to a static set of ETF vertices” (NCT) to “continually redefining and expanding the ETF structure itself” in a way that remains compatible with neural-collapse geometry and the stability–plasticity requirements of continual learning.
>
> Importantly, this progressive construction also leads to a different geometric regime from NCT. For NCT, the pairwise inner product between any two normalized ETF vertices is $−1/(K\_{global}−1)$, which is very close to 0 when $K\_{global}​$ is large (e.g., −1/999 for $K\_{global}​$=1000). This means early-task classes are packed with relatively small angular separation. In ProNC, at Task 1 with $K\_1$=10 classes, the ETF is built only for these 10 classes. The inner product between any two vertices is −1/($K\_1$−1)=−1/9, which corresponds to a larger angle (lower similarity). Intuitively and as corroborated by our results, this larger angle between different vertices makes it easier to separate classes during early tasks.
>
> Beyond the high-level idea of “dynamic ETF expansion,” there are additional algorithmic and theoretical components that are new rather than direct adaptations of NCT. We formulate how to expand the ETF in an appropriate way, which is non-trivial: the expansion must respect the geometric constraints of ETFs (equiangularity, equal norms, etc.) and the incremental arrival of new classes. This requires a new construction, not present in NCT, for generating new ETF targets that remain coherent with the existing ones as tasks progress.
>
> This, in turn, leads to a new training framework where, at each task, features are aligned to an updated, higher-dimensional ETF structure rather than to a fixed set of targets. Designing this update rule and integrating it into the continual learning training loop is itself an algorithmic contribution beyond NCT’s fixed-ETF setting.
>
> By addressing the limitations of using a fixed ETF in continual learning (as discussed in the paper), ProNC demonstrates systematically improved performance over prior neural-collapse-based methods. This improvement is viewed as empirical evidence that the proposed dynamic ETF expansion framework unlocks the additional potential of neural collapse for CL, beyond what is captured by NCT.
>
> Q2: Different performance of ER in the GPM paper
>
> A2: Thank you for raising this point. Our results are obtained under a different protocol than the one used in GPM, so they are not directly comparable. Most importantly, the memory budget is much smaller in our setting: on Seq-CIFAR100, we report results with 200 and 500 exemplars in total, whereas GPM uses a 2000 memory buffer for Split CIFAR-100. A larger buffer is well known to substantially improve the performance of rehearsal-based methods by reducing forgetting, resulting in a significant performance improvement. Besides the memory size, we only trained 50 epochs for CIFAR100, and in GPM, they trained maximally 200 epochs for CIFAR100, which could also cause the performance gap.

---

> ### Author Response · Authors · 2025-11-25
>
> Q3: Losses novelty compared with the preliminary works (Yang et al., 2023a,b)
>
> A3: We agree that the losses in Section 3.2 closely resemble those of NCT, and this is intentional. Our goal is not to propose a new neural-collapse loss, but to study how neural collapse can be exploited more effectively in continual learning through the design of the ETF targets and their progressive expansion.
>
> To verify that our framework does not rely on a particular loss form, we report in the appendix an ablation where we replace the original loss with a simple L2-norm loss between features and ETF targets. The results change only marginally, indicating that any reasonable loss that minimizes the distance between two vectors works similarly well in our setting. In other words, what matters is enforcing alignment to the prescribed targets, rather than the specific choice of distance metric.
>
> Therefore, the main novelty of ProNC does not lie in Section 3.2’s loss expression itself—which largely follows standard neural-collapse formulations such as NCT—but in how we construct and update the ETF targets over time (Section 3.1) and integrate this progressive ETF expansion into the continual learning protocol. We will clarify this focus in the revision and explicitly state that we adopt a conventional NC-style loss on purpose, to isolate and highlight the contribution of our target design.
>
> We thank the reviewer again for the helpful comments and suggestions for our work.

---

> > ### Comment · Reviewer_cHhJ · 2025-11-26
> >
> > Thank you for the clarification and the detailed responses.
> >
> > While some of my earlier questions (especially Q2) have been addressed, the core novelty concern remains largely unresolved. As the authors themselves acknowledge, the loss formulation—which constitutes a substantial portion of the technical contribution in the original submission—remains nearly identical to prior work, regardless of whether this was intentional. The idea of dynamically expanding the ETF is interesting, but on its own does not appear sufficient to constitute a complete and standalone research contribution.

---

> > > ### Author Response · Authors · 2025-12-03
> > >
> > > Thank you for your response and confirming our clarifications. Regarding your concerns about the novelty, we have the following response:
> > >
> > > We respectfully clarify that the core contribution of ProNC is not the loss function, but the algorithmic framework that constructs and expands an ETF geometry across incremental tasks. Your concern over the loss function overlooks the central novelty of our method.
> > >
> > > Specifically, ProNC introduces a two-stage geometric mechanism:
> > >  (1) after training the base task, we derive the base ETF directly from the empirical class means, and
> > >  (2) for each incremental task, we expand the ETF by extending the orthogonal matrix that defines the simplex structure,
> > > thereby preserving angular separation and maintaining a stable feature geometry throughout continual learning.
> > >
> > > This geometric design—not the loss form—is the fundamental innovation. In fact, as we prove in the paper, any loss that minimizes the distance between two vectors is sufficient for our framework, meaning the loss is merely an instantiation and not essential to the algorithm’s novelty. This theoretical result deliberately decouples ProNC from any specific loss, demonstrating that the method is general, modular, and not tied to a particular objective function.
> > >
> > > Because of this generality, ProNC naturally serves as a regularization module that can be added to existing rehearsal-based methods, which we validate experimentally. The improvement we observe when integrating ProNC into other methods further confirms that the strength of our approach lies in the ETF construction and expansion strategy, not in a particular choice of distance-based loss.

---

### Author Response · Authors · 2025-11-25
**Summary of Revision Paper Updates**

We sincerely thank all reviewers for their thorough assessments and insightful comments. We have carefully revised the manuscript to address your concerns. The major updates in the revision are summarized below:

1. Enhanced Clarity: We have rewritten Section 3.1, Part 2 to improve the readability and clarity of the explanation regarding "ETF expansion prior to new task learning."

2. Expanded Empirical Evaluation:

    Comprehensive Benchmarking: We incorporated the LODE results into Table 1 and introduced Table 2 to demonstrate the efficacy of our approach in buffer-free settings.

    Versatility: Table 3 now includes results showing the performance gains achieved by integrating LODE as a regularization term into existing rehearsal-based frameworks (e.g., ER and DER++).

3. Novelty & Flexibility: We have expanded the discussion to highlight the distinct benefits of our method and included an analysis demonstrating the flexibility of the different loss functions (Table 4).

4. Baseline Reproducibility: We explicitly clarified our experimental setup to explain the performance discrepancies observed in certain baselines compared to their original literature, ensuring a fair and transparent comparison.

We look forward to further discussion and appreciate your time.

---

### Author Response · Authors · 2025-12-03
**Summary of Revision and Response to Reviewers**

Dear Area Chair,
We sincerely thank you for overseeing the review process and for handling our paper. We are grateful for the constructive feedback provided by the reviewers.

In this revision, we have clarified our contributions and added significant new experimental evidence to address the raised concerns. Below is a summary of the key updates and clarifications.
1. Summary of Novelty
Our paper introduces ProNC, a dynamic ETF expansion framework for continual learning. ProNC fundamentally extends the neural collapse phenomenon beyond the fixed Simplex Equiangular Tight Frame (ETF) used in prior work (e.g., NCT).
The Innovation: We develop a principled mechanism to expand the ETF geometry as new classes arrive.
The Benefit: Unlike static approaches, our method ensures that strict ETF properties (equiangularity, equal norms) are preserved dynamically. This allows features to align with a task-adaptive structure rather than a static target, providing better-conditioned geometry and consistently stronger performance throughout incremental training.
2. New Experiments (Response to Reviewers)
We have added extensive experiments to the revision to demonstrate robustness and generality:
Updated Baselines (Main Table): We have incorporated SOTA methods (STAR, LODE) and updated the best performance metrics, showing that ProNC outperforms existing methods in most scenarios.
Generality (Table 3): We added experiments using ProNC as a regularization term for other rehearsal-based methods, demonstrating that our geometric constraints can boost the performance of other frameworks.
Flexibility (Table 4): We included results using alternative loss functions to demonstrate that ProNC is not tied to a specific objective.
Impact of ETF Size (Figure 1): We empirically demonstrate that pre-defining a large ETF for unknown-sized datasets degrades performance, validating the need for our dynamic approach.
3. Clarifications on Specific Concerns

      - **On the Loss Function (Reviewer cHhJ):** Reviewer cHhJ noted similarities between our loss and that of NCT. We clarify that the core novelty of ProNC is the algorithmic framework for constructing and expanding ETF geometry, not the loss function itself. As proven in our paper, our framework is loss-agnostic: any loss minimizing the distance between vectors suffices. The loss used is merely an instantiation, and our new ablation studies (Table 4) confirm the framework's flexibility.

      - **On the Impact of Pre-defined ETF Size (Reviewer fz6X):** Reviewer fz6X suggested that initializing a large number of classes (large $K_{global​}$) does not hinder performance. We respectfully disagree based on both geometric intuition and empirical results:
         * Geometric constraint: In a fixed ETF with a large $K_{global​}$​, the pairwise inner product is $−1/(K_{global}​−1$). For a large $K$ (e.g., 1000), this value is near zero, meaning class separation is weak.
         * ProNC Advantage: By constructing the ETF dynamically (e.g., $K_1​=10$ for Task 1), the inner product is −1/9. This larger angular separation creates a better-conditioned geometry for early tasks.
         * Evidence: Our new experiments (Figure 1) confirm that performance drops as the pre-defined ETF size increases, supporting the necessity of our dynamic expansion strategy.

      - **On Baseline Performance Discrepancies (Reviewers cHhJ & fz6X):** We noted that our baseline results differ from the original papers due to the standardization of training epochs for fair comparison. While methods like Co2L and CILA originally used up to 500 epochs, we standardized comparisons to 50 epochs for CIFAR-10/100 and 100 epochs for Tiny-ImageNet to ensure a consistent experimental setup across all methods.

We believe this revision significantly improves the paper's clarity and robustness. Thank you again for your time and consideration.

Best regards,

The authors

---

### Meta-Review · Area_Chair_WYgP · 2025-12-11

**Summary:**

## Reviewer CHhJ

This reviewer was concerned about the novelty of this work with respect to NCT. The reviewer also had some concerns about the numbers reported for the baselines, which was resolved by the authors.

## Reviewer hVBn

Was concerned about the degradation of orthogonality with more tasks, which the authors correctly resolved. They were also concerned about the focus on task-aware CL without considering other setups, which was partially addressed by the authors with a justification.


## Reviewer fz6X

This reviewer was also concerned about the similarities with NCT, lower baseline performance, and asked the authors to include more baselines.

**Reviewer Concerns:**

## Reviewer CHhJ

The concerns about baseline numbers were addressed by the authors. The authors also clarified that while NCT and their method share the same loss, they contribute with a new algorithmic framework.

## Reviewer hVBn

The orthogonality degradation concerns were correctly addressed. The concern about the focus on task-aware CL without considering other setups is not addressed but properly justified.

## Reviewer fz6X

The baseline performance concerns were addressed, and the authors included new baselines:

> New Experiments (Response to Reviewers) We have added extensive experiments to the revision to demonstrate robustness and generality: Updated Baselines (Main Table): We have incorporated SOTA methods (STAR, LODE) and updated the best performance metrics, showing that ProNC outperforms existing methods in most scenarios. Generality (Table 3): We added experiments using ProNC as a regularization term for other rehearsal-based methods, demonstrating that our geometric constraints can boost the performance of other frameworks. Flexibility (Table 4): We included results using alternative loss functions to demonstrate that ProNC is not tied to a specific objective. Impact of ETF Size (Figure 1): We empirically demonstrate that pre-defining a large ETF for unknown-sized datasets degrades performance, validating the need for our dynamic approach.

**Reviewer Scores:**

* Reviewer CHhJ
  * Original Score: 4
  * Updated Score: 6
  * Reason: Although the authors addressed concerns on baseline performance, the reviewer might have remained concerned about the significance of the proposed method in light of previous methods like NCT but I believe the authors did a good job clarifying the novelty of their work.
* Reviewer hVBn
  * Original Score: 6
  * Updated Score: 6
  * Reason: This score is already the highest, and the justification for only addressing task-aware CL might not have addressed the reviewer concern.
* Reviewer fz6X
  * Original Score: 2
  * Updated Score: 4
  * Reason: This reviewer is also concerned about novelty but the authors did a great effort to add more baselines. In fact they add ProNC on top of xDER and it improves.

## Decision
Accept. While there is some concern about the novelty, this work addresses limitations of existing works, providing a method that is more flexible and performs well across different evaluations.

---

### Decision · Program_Chairs · 2026-01-26

Accept (Poster)